# Dock1 functions in Schwann cells to regulate development, maintenance, and repair

Ryan A. Doan[1] and Kelly R. Monk[1]

**Schwann cells, the myelinating glia of the peripheral nervous system (PNS), are critical for myelin development, maintenance, and repair. Rac1 is a known regulator of radial sorting, a key step in developmental myelination. Previously, in zebrafish, we showed that the loss of Dock1, a Rac1-specific guanine nucleotide exchange factor, resulted in delayed peripheral myelination during development. Here, we demonstrate that Dock1 is necessary for myelin maintenance and remyelination after injury in adult zebrafish. Furthermore, Dock1 performs an evolutionarily conserved role in mice, functioning cell autonomously in Schwann cells to regulate the development, maintenance, and repair of peripheral myelin. Pharmacological and genetic manipulation of Rac1 in larval zebrafish, along with the analysis of active Rac1 levels in developing *Dock1* mutant mouse nerves, revealed an interaction between these two proteins. We propose that the interplay between Dock1 and Rac1 signaling in Schwann cells is required to establish, maintain, and facilitate repair and remyelination within the PNS.**

## Introduction

Myelin, the lipid-rich multilamellar sheath that surrounds and insulates axons, plays a critical role in the vertebrate nervous system, enabling rapid transmission of nerve impulses (Jessen and Mirsky, 2005). In the peripheral nervous system (PNS), myelin is synthesized by Schwann cells (SCs), with each mature SC myelinating a single axonal segment (Monk et al., 2015). Derived from the neural crest, SCs progress through developmental stages delineated by the expression of specific genes and marked by significant morphological transformations (Ackerman and Monk, 2016). SC precursors undergo extensive longitudinal migration along pathfinding peripheral axons and subsequently differentiate into immature SCs, which perform a specialized function known as radial sorting. During this process, an immature SC projects extensions into a bundle of axons and selectively identifies axon segments to myelinate (Feltri et al., 2016). Following radial sorting, immature SCs that selected larger caliber axons enter a pro-myelinating state, enveloping and myelinating the chosen axon segment. Smaller caliber axons that do not become myelinated associate with Remak SCs and form clusters of unmyelinated axons known as Remak bundles (Harty and Monk, 2017; Herbert and Monk, 2017). Proper regulation of SC homeostasis is required beyond development, as it is necessary to maintain myelin and function in repair and remyelination in cases of injury and disease (Bremer et al., 2011; Jessen and Mirsky, 2016, 2019). While a large body of work has shed light on the multifaceted functions SCs play throughout life (Taveggia and Feltri, 2022), a

complete understanding of the signaling involved at each stage remains incompletely defined and represents a critical area for further exploration.

Work from our lab previously showed that Dock1, an evolutionarily conserved guanine nucleotide exchange factor (GEF), is required for timely radial sorting and developmental PNS myelination in zebrafish (Cunningham et al., 2018). Dock1 belongs to an 11-member family of Dock proteins, related in their ability to activate Rac1, fellow Rho family member Cdc42, or a combination of both (Côté and Vuori, 2002). GEFs play a direct role in activating Rho family GTPases in reaction to various extracellular signals and activity, enabling them to function as regulators of the cytoskeletal dynamics that underpin numerous cellular processes, ranging from migration, morphological changes, and phagocytosis (Côté and Vuori, 2002, 2007; Hasegawa et al., 1996; Laurin et al., 2008; Rossman et al., 2005; Ruiz-Lafuente et al., 2015; Ziegenfuss et al., 2012). Additional work has begun to characterize the importance of several GEFs, including members of the Dock family, in regulating SC development and function (Miyamoto et al., 2016; Pasten et al., 2015; Yamauchi et al., 2008, 2011). Dock1 specifically regulates the Rho-GTPase Rac1, an essential mediator of SC development that governs shape changes via regulation of the actin cytoskeleton (Kiyokawa et al., 1998; Nodari et al., 2007). During SC development, temporally varied levels of Rac1 sequentially control migration, commencement of radial sorting, and myelination. In a mouse model with SC-specific deletion of *Rac1*, SCs

[1]The Vollum Institute, Oregon Health & Science University, Portland, OR, USA.

Correspondence to Kelly R. Monk: monk@ohsu.edu.



in developing sciatic nerves showed evidence of delayed radial sorting along with abnormal SC cytoplasmic extensions, ultimately resulting in severely delayed myelination (Benninger et al., 2007; Guo et al., 2012; Nodari et al., 2007). The function of Dock1 in the PNS is an emerging area of interest, and while it has been demonstrated to be required for proper PNS development in zebrafish (Cunningham et al., 2018), the underlying mechanisms of its function and roles in myelin maintenance, repair, and remyelination following injury have not been examined. Furthermore, *Dock1*'s high expression in the developing mouse PNS (Gerber et al., 2021) underscores its potential significance, yet its specific function in mammalian SCs is unknown.

In this study, we employ zebrafish and mouse models to expand our knowledge of how Dock1 functions in the PNS. Our data reveal that Dock1 is instrumental for development but is dispensable for myelin maintenance into early adulthood in both zebrafish and mice. We show that aged animals in both species rely on Dock1 for the long-term maintenance of myelin integrity, with mature animals manifesting numerous aberrant myelin phenotypes. Moreover, we identify a critical function for Dock1 in the remyelination of axons after peripheral injury. Manipulating Rac1 levels during zebrafish development reveals an interaction with *dock1*, which influences developmental myelination. In addition, we demonstrate that mice with SC-specific deletion of *Dock1* have lower levels of active Rac1 in the developing sciatic nerve. Collectively, these findings illuminate Dock1's complex and evolutionarily conserved role in SCs, where it regulates myelin development, homeostasis, and repair. Understanding the interplay between Dock1 and Rac1 may provide new insights into the pathways controlling myelin formation and maintenance, offering novel avenues for treating conditions that impact the PNS.

## Results

### Dock1 functions in myelin maintenance in aged adult zebrafish
Our previously published work identified Dock1 as a regulator of developmental PNS myelination in zebrafish (Cunningham et al., 2018). Given that many genes required for myelin development are also necessary for myelin maintenance (Decker et al., 2006), we wanted to know if Dock1 played a role in myelin homeostasis into adulthood. To this end, we analyzed zebrafish maxillary barbels (ZMBs) using the previously described *dock1*[stll45] loss of function mutant (MUT) zebrafish line. *stll45*, the allele designation of the *dock1* MUT we previously identified in a forward genetic screen, represents an early stop codon in the Rac1-binding domain of Dock1. *dock1*[stll45/+] heterozygous (HET) MUTs do not show any myelin phenotypes, while *dock1*[stll45/stll45] homozygous MUTs exhibit delayed developmental myelination and have evidence of delayed radial sorting (Cunningham et al., 2018). Maxillary barbels are innervated sensory organs found in fish, reptiles, and amphibians (Winokur, 1982). Zebrafish develop paired ZMBs at ~1 mo of age (LeClair and Topczewski, 2009). They contain a variety of structures, including taste buds, goblet cells, and a population of pure sensory nerves branching from cranial nerve VII (LeClair and Topczewski,

2009, 2010; Moore et al., 2012). We performed ultrastructural analyses of ZMBs from 4-mo-old and 12-mo-old WT, *dock1*[stll45/+] HET, and *dock1*[stll45/stll45] homozygous MUT animals by transmission EM (TEM). At 4 mo, we observed no changes in either the number of myelinated axons or in g-ratios, nor did we note any obvious myelin defects in HET or homozygous MUT ZMBs compared with WT controls (Fig. 1, A–C and I; and Fig S1, A–D). At 12 mo, however, we found there was a significant increase in the percentage of abnormally myelinated axon profiles in homozygous MUTs compared with WT and HET controls (Fig. 1, D–I; WT 12 mo old = 5.54%, MUT 12 mo old = 7.84%, P = 0.0002). These findings suggest that Dock1 is dispensable during early adulthood but necessary for long-term myelin maintenance in zebrafish.

### Remyelination following injury is significantly impaired in *dock1* MUT zebrafish
To further our understanding of the role of Dock1 in the developed PNS, we examined its role in remyelination following injury in zebrafish. The ZMB can regrow after amputation, and axons within nerves of the regenerating ZMB are remyelinated as the appendage regrows, with myelin reaching around 85% of its original thickness 4 wk following transection (Moore et al., 2012). We therefore cut and removed the left ZMB from 3-mo-old animals (a time point when myelin maintenance defects are not yet apparent) and allowed recovery for 4 wk. The right uncut ZMB served as an internal uninjured control for each animal. After 4 wk, ZMBs from both sides were removed, processed for TEM, and the nerves were examined. The regenerated ZMBs of the WT, HET (data not shown), and homozygous *dock1* MUTs were similar in appearance and had both regrown to ~90% of their original length (Fig. S2, A–D), suggesting that Dock1 is not required for gross ZMB regeneration. We observed, however, a profound loss in the number of myelinated axons in the regenerated ZMBs of homozygous *dock1* MUTs compared with WT (Fig. 2, A–E; WT control = 19.85, WT regenerated = 16.87, MUT control = 16.82, MUT regenerated = 4.80, P = <0.0001). Moreover, the myelin in the homozygous MUTs was significantly thinner than in WT, as analyzed by g-ratio (Fig. 2 F; WT regenerated = 0.6747, MUT regenerated = 0.8283, P = <0.0001). SC nuclei number, total axon number, and axon size were quantified by examining TEM micrographs, and we observed no differences in these parameters between regenerated WT and *dock1* homozygous MUT ZMBs (Fig. S2, E–G). These results suggest a crucial function for Dock1 in regulating remyelination of the PNS following nerve injury in zebrafish.

### Dock1 functions cell autonomously in SCs to regulate myelination
Our work in global zebrafish *dock1* MUTs demonstrates a vital role for Dock1 in regulating PNS myelin, from development to maintenance and repair. We hypothesized that loss of Dock1 function, specifically in SCs, is responsible for the phenotypes we observe in zebrafish for three reasons: (1) *Dock1* is highly expressed in developing SCs (Gerber et al., 2021); (2) The known link between Dock1 and Rac1 signaling (Côté and Vuori, 2007); (3) The importance of Rac1 signaling in SCs (Benninger et al., 2007; Nodari et al., 2007). To test this theory and simultaneously

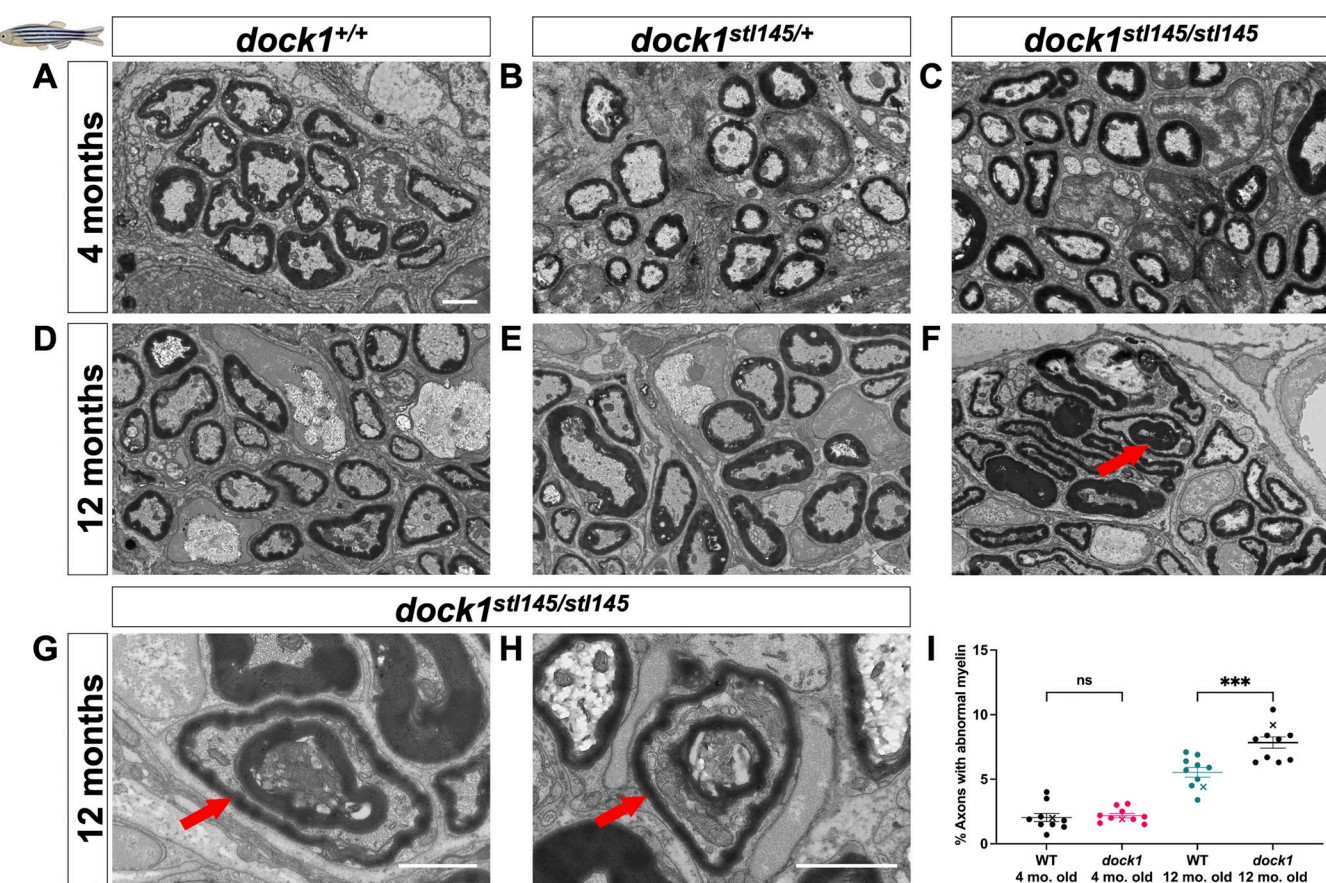

**Figure 1. Age-dependent myelin maintenance defects are present in *dock1* MUTs. (A–C)** TEM of cross-sections of ZMBs from 4-mo-old WT *dock1*[+/+], HET *dock1*[stl145/+], and homozygous *dock1*[stl145/stl145] MUT zebrafish. **(D–F)** TEM micrographs of ZMBs from 12-mo-old WT *dock1*[+/+], HET *dock1*[stl145/+], and homozygous *dock1*[stl145/stl145] MUT zebrafish, with homozygous MUTs exhibiting myelin outfoldings (red arrow). **(G and H)** Higher-magnification TEM micrographs of ZMBs from 12-mo-old homozygous *dock1*[stl145/stl145] MUTs showing abnormally myelinated axons (red arrows), features rarely seen in WT *dock1*[+/+] or HET *dock1*[stl145/+] animals. **(I)** Quantification of the percent of axons with abnormal myelin profiles, observed by TEM, in WT *dock1*[+/+] versus homozygous *dock1*[stl145/stl145] MUTs at 4 and 12 mo old, *n* = 10 (WT *dock1*[+/+] 4 mo old), 10 (*dock1*[stl145/stl145] MUT 4 mo old), 10 (WT *dock1*[+/+] 12 mo old), and 10 (*dock1*[stl145/stl145] MUT 12 mo old). Here, and in all figures, the X symbol in the graph denotes a data point corresponding to the representative image shown. **(A–H)** Scale bar = 1 μm. **(I)** Two-way ANOVA with Tukey's multiple comparisons test. ***P < 0.001; ns, not significant.

determine if the function of Dock1 is evolutionarily conserved in mammals, we generated SC-specific *Dock1* conditional knockout (cKO) mice by crossing validated *Dock1*[fl/fl] mice (Laurin et al., 2013) with the well-characterized *Dhh*[Cre] mouse line (Jaegle et al., 2003) to drive recombination in SC precursors at approximately embryonic day 12.5. Western blotting revealed a ~70% reduction in Dock1 protein levels in the sciatic nerve of *Dhh*[(Cre+)];*Dock1*[fl/fl] (cKO) mice compared with littermate controls (Fig. 3 A). We first examined the sciatic nerves of animals on postnatal day (P)3, when radial sorting is actively underway (Ackerman and Monk, 2016). Ultrastructural analyses by TEM revealed that *Dock1* cKO animals had thinner myelin than their littermate controls (Fig. 3, B–D; control = 0.7228, cKO = 0.7759, P = <0.0001). To determine if this was due to a broader developmental defect in the SCs or the nerve itself, we examined TEM images and quantified the number of SC nuclei, total axons, and myelinated axons. We found no significant differences between groups (Fig. S3, A–C). Upon closer examination of higher-magnification TEM micrographs, we noticed that the SCs in the MUT animals exhibited additional defects. These included

elaborate cytoplasmic protrusions extending from MUT SCs (Fig. 3 E) and evidence of basal lamina trails in regions devoid of SC cytoplasm (Fig. 3 F), suggesting that unstable SC process extensions had been made and retracted (Benninger et al., 2007; Nodari et al., 2007). To determine if these defects persisted throughout development, we examined animals at P28, when radial sorting was complete and most myelin was established (Ackerman et al., 2018). Interestingly, at P28, *Dock1* cKO animals appear indistinguishable from littermate controls (Fig. S3, D and E). There was no difference in the myelinated axon number (Fig. S3 F), and the increased g-ratio observed in the MUTs at P3 had resolved (Fig. S3 G). These findings parallel our observations in zebrafish and reveal that Dock1 is an evolutionarily conserved regulator of developmental myelination, functioning cell autonomously in SCs.

### *Dock1* SC-specific knockout mice show age-associated myelin abnormalities

Proper maintenance of SCs in the developed PNS is essential for these cells to support the physiological health of the adult. When

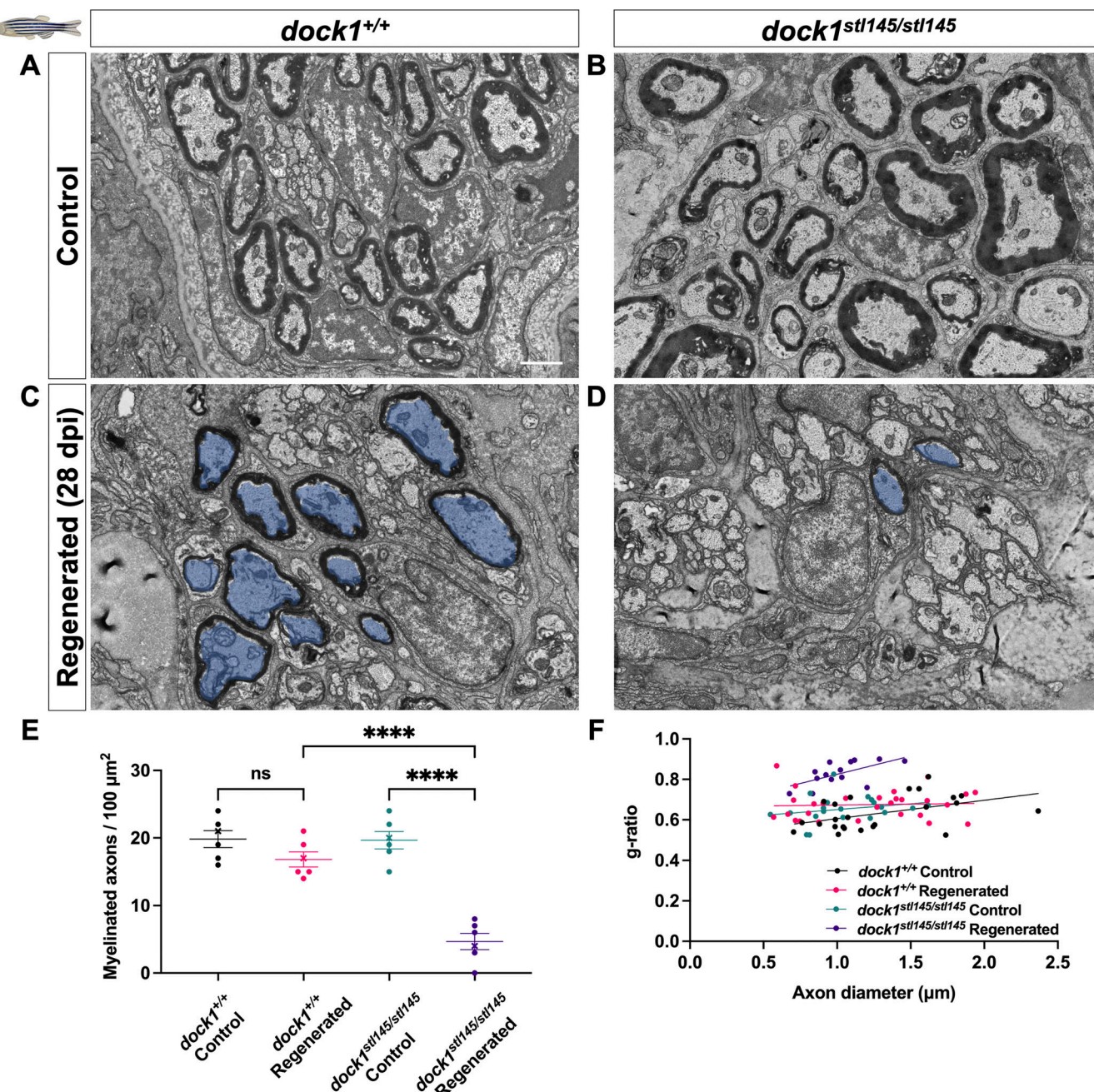

Figure 2. **Remyelination following nerve injury is significantly reduced in *dock1* MUTs. (A and B)** TEM micrographs of control ZMBs from 4-mo-old WT *dock1⁺/⁺* and MUT *dock1ˢᵗˡ¹⁴⁵/ˢᵗˡ¹⁴⁵* zebrafish. **(C and D)** TEM micrographs of WT *dock1⁺/⁺* and MUT *dock1ˢᵗˡ¹⁴⁵/ˢᵗˡ¹⁴⁵* zebrafish showing regeneration and remyelination after transection, with remyelinated axons pseudocolored in blue. **(E)** Quantification of the number of myelinated axons in the ZMBs per 100 μm², n = 6 (WT *dock1⁺/⁺* control), 6 (WT *dock1⁺/⁺* regenerated), 6 (*dock1ˢᵗˡ¹⁴⁵/ˢᵗˡ¹⁴⁵* MUT control), and 6 (*dock1ˢᵗˡ¹⁴⁵/ˢᵗˡ¹⁴⁵* MUT regenerated). **(F)** Quantification of the g-ratio as it relates to the axon caliber of the remyelinated axons in the regenerated ZMBs, 28 days after transection, n = 6 (WT *dock1⁺/⁺* control), 6 (WT *dock1⁺/⁺* regenerated), 6 (*dock1ˢᵗˡ¹⁴⁵/ˢᵗˡ¹⁴⁵* MUT control), and 6 (*dock1ˢᵗˡ¹⁴⁵/ˢᵗˡ¹⁴⁵* MUT regenerated). **(A–D)** Scale bar = 1 μm. **(E)** Two-way ANOVA with Tukey's multiple comparisons test. ****P < 0.0001; ns, not significant.

mature SC homeostasis is disrupted, it can present with various abnormalities, including muscle atrophy, decreased nerve conduction velocities, and sensory loss (Verdú et al., 2000). Several MUTs with abnormal SC development are often accompanied by lifelong myelin defects (Bremer et al., 2011; Decker et al., 2006). In contrast, the delayed radial sorting and developmental hypomyelination seen in our *Dock1* MUTs resolved as early as P28.

Some MUTs, such as *Gpr56/Adgrg1*, have a similar pattern of developmental SC defects that recover by early adulthood but show myelin abnormalities with age (Ackerman et al., 2018). To determine if this was the case for Dock1, we performed ultrastructural analyses of mouse sciatic nerves at 12 mo of age. TEM revealed numerous myelin abnormalities in the 12-mo-old *Dock1* cKO MUTs compared with their younger P28 counterparts and

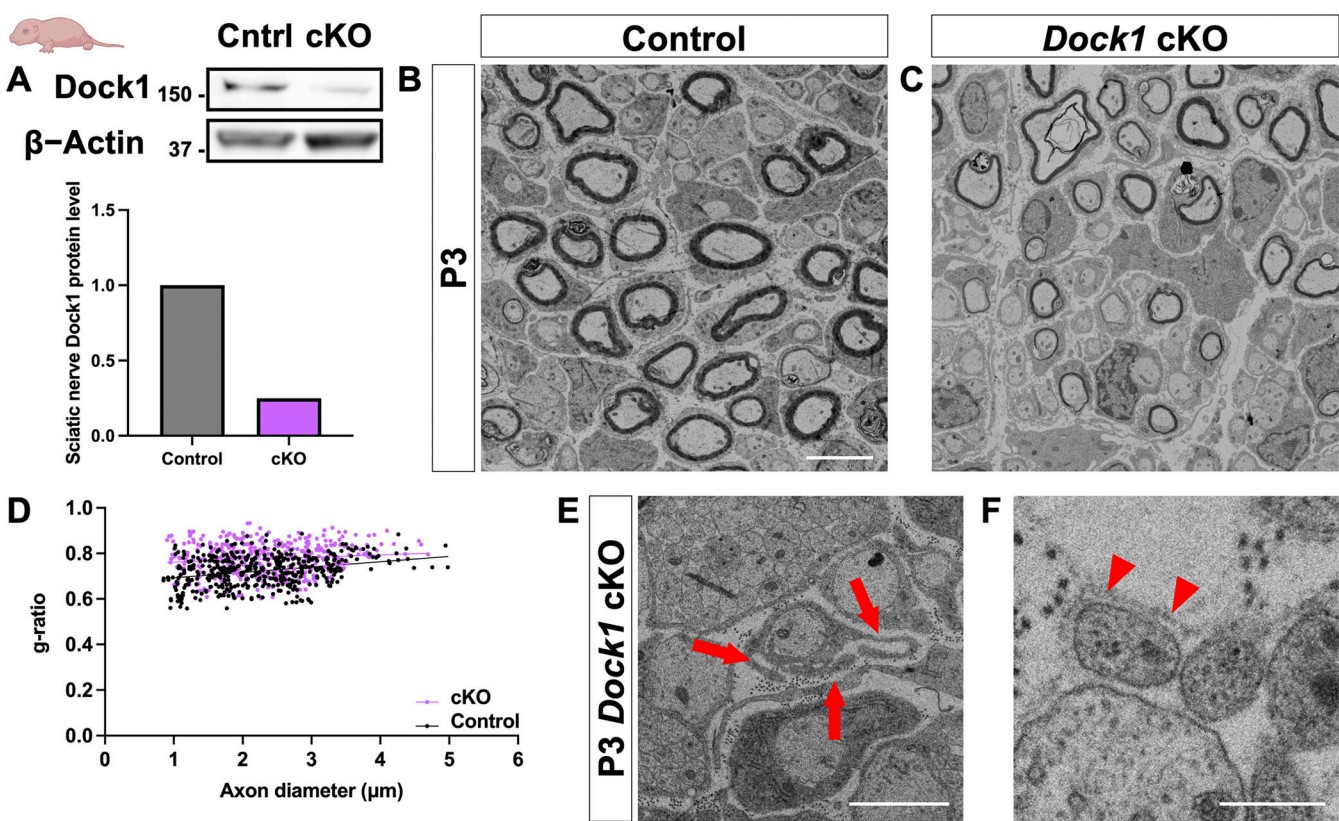

**Figure 3.** **SC-specific *Dock1* MUTs present with multiple defects in peripheral nerves. (A)** Western blot (kD) showing sciatic nerve Dock1 and β-actin protein levels from control and *Dock1* cKO animals and quantification of normalized protein levels. **(B and C)** TEM micrographs of sciatic nerves from *Dhh^{Cre+}*; *Dock1^{+/+}* control and littermate *Dhh^{Cre+};Dock1^{fl/fl}* cKO mice at P3. **(D)** Quantification of the g-ratio as it relates to axon caliber, *n* = 6 mice, 4 images per nerve (WT); 6 mice, 4 images per nerve (cKO). **(E)** *Dock1* cKO MUT SCs display abnormal cytoplasmic protrusions that extend in multiple directions (red arrows). **(F)** Trails of basal lamina found in *Dock1* cKO MUTs are observed in regions devoid of SC cytoplasm (red arrowheads). **(B and C)** Scale bar = 4 μm. **(E and F)** Scale bar = 1 μm. Source data are available for this figure: SourceData F3.

age-matched littermates, including abnormal myelinated fibers and Remak defects (Fig. 4, A and B). We saw signs of degenerating myelin sheaths and accumulated axonal debris (Fig. 4, C–E), as well as regeneration clusters (Fig. 4 F) and myelin outfoldings (Fig. 4 G). Although control mice also showed some defects with age, these abnormalities were significantly more prevalent in MUTs (Fig. 4 H; control 12 mo = 4.02% of axons with abnormal myelin profiles, cKO 12 mo = 12.41% of axons with abnormal myelin profiles, P = 0.0027). These findings indicate that Dock1 is required in mouse SCs for long-term myelin maintenance and axonal health and align with our observations in zebrafish, where myelin is normal in early adulthood, but defects arise with age.

## Regeneration and remyelination are impaired in inducible *Dock1* SC-specific knockout mice

To help integrate the findings from our ZMB injury model and the cell-autonomous role of Dock1 in SCs, we examined its importance in mammalian remyelination. To assess this, we used the well-characterized *Plp^{CreERT2}* mouse line (Leone et al., 2003) to generate an inducible cKO (icKO) mouse. This allowed us to disrupt *Dock1* in mature SCs, leaving it functional during development. To assess repair after injury, we performed sciatic nerve transections, where the role of the SCs in regeneration

and remyelination has been well described (Jessen and Mirsky, 2016). We transected the sciatic nerves of 3-mo-old tamoxifen-injected *Plp^{Cre+};Dock1^{fl/fl}* (icKO) mice and corn oil–injected *Plp^{Cre+}*; *Dock1^{fl/fl}* (control) mice 4 wk following the final tamoxifen injection, allowing sufficient time for recombination (Fig. S4 A) (Leone et al., 2003; Mogha et al., 2016). Following nerve transection, a bridge rapidly forms, and the nerve regenerates, making it difficult to see the injury site without resorting to immunostaining (Cattin and Lloyd, 2016; Dun and Parkinson, 2015). To ensure we examined nerves at the same distance from the cut site when we performed TEM, we quickly crushed the nerve with forceps coated in activated charcoal to mark the cut site before transection. We examined and analyzed the nerves at 14 and 25 days postinjury (dpi), time points that allow us to assess the clearance of debris associated with degenerating axons and also remyelination, respectively (Wang et al., 2023). Western blotting revealed a ~40% reduction in Dock1 protein levels in the sciatic nerve of our tamoxifen-injected *Plp^{Cre+}*; *Dock1^{fl/fl}* mice compared with corn oil–injected controls (Fig. S4 B). When we examined the uninjured nerves of the 4-mo-old icKO mice, we saw that the myelin abnormalities observed at 12 mo in the cKO mice had yet to arise (Fig. S4, C and D). At 14 dpi, icKO mice had significantly more macrophages containing intact myelin cylinders (George and Griffin, 1994) than controls (Fig. 5,

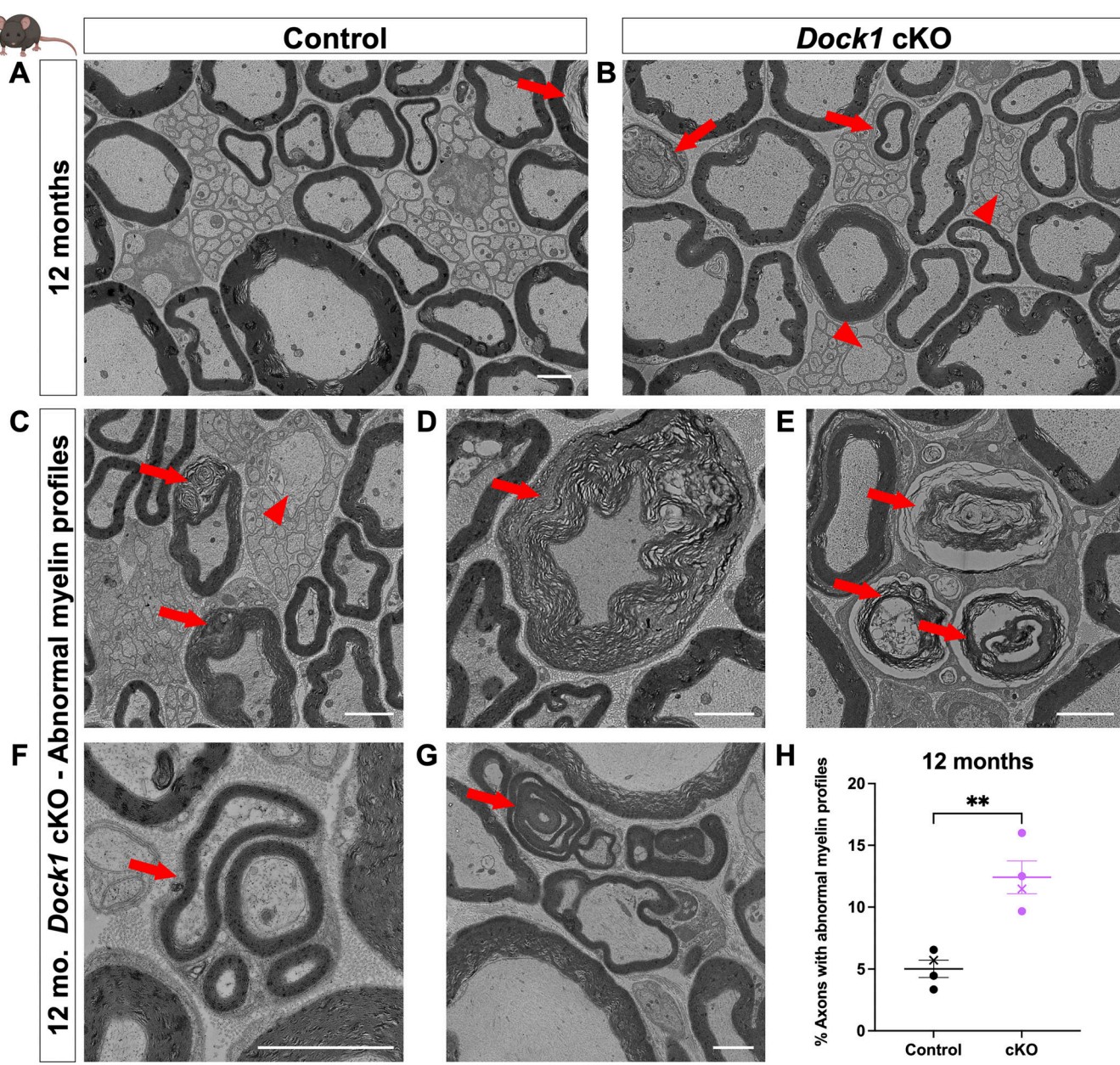

Figure 4. **Myelin maintenance defects arise and accumulate with age in *Dock1* cKO mice. (A and B)** TEM micrographs of control and MUT nerves at 12 mo with MUTs containing aberrant myelin (red arrows) and Remak (red arrowheads) phenotypes. The following higher-magnification TEM micrographs are from 12-mo-old *Dock1* cKO nerves: **(C)** Degenerating myelin (red arrows) and a large-caliber (>1 µm) axon in a Remak bundle (red arrowhead); **(D)** Disorganized myelin sheath (red arrow); **(E)** Accumulations of myelin debris and degenerating sheaths (red arrows). **(F)** Regeneration clusters (red arrow). **(G)** Myelin outfoldings and abnormal wrapping (red arrow). **(H)** The percentage of axons with abnormalities, *n* = 4 mice, 4 images per nerve (control); 4 mice, 4 images per nerve (*Dock1* cKO). Here, and in all figures, the X symbol in the graph denotes a data point corresponding to the representative image shown. **(A–G)** Scale bar = 2 µm. **(H)** Unpaired *t* test with Welch's correction. **P < 0.01.

A–C; control = 5.208, icKO = 19.98, P = 0.0376), indicative of delayed debris clearance. In addition, there were significantly more foamy macrophages present in icKO mice (Fig. 5, A, B, and D; control = 0.9286, icKO = 2.071, P = 0.0434) and increased numbers of presumptive motile macrophages, distinguished by irregular shape, cytoplasmic processes, and a lack of phagocytosed debris (Fry et al., 2007) (Fig. 5, A, B, and E; control = 0.5025, icKO = 1.643, P = 0.0337), altogether suggesting delayed and ongoing macrophage recruitment. icKO mice showed a

trend toward having fewer Bands of Büngner than controls (Fig. 5 F). These findings raise the possibility of icKO mice having delayed debris clearance, resulting in an environment less conducive for axon regeneration at 14 dpi. By 25 dpi, control animals showed axons of various calibers that had begun to remyelinate (Fig. 5 G). In contrast, icKO nerves had significantly fewer remyelinated axons (Fig. 5, H and I; control = 6.69, icKO = 2.21, P = 0.0021), and those present had higher g-ratios (Fig. 5 J; control = 0.7329, icKO = 0.8172, P = <0.0001),

Figure 5. **Remyelination is delayed following sciatic nerve transection in *Dock1* icKO mice. (A and B)** TEM micrographs of sciatic nerves from control-injected (control) and tamoxifen-injected *Plp^Cre+;Dock1^fl/fl* mice (icKO) 14 days after transection. Quantification reveals that icKO mice show **(C)** higher numbers of macrophages containing intact myelin cylinders (red arrow), **(D)** foamy macrophages (red arrowheads), and **(E)** motile macrophages (asterisks). **(F)** There was a trend in the icKO animals toward having fewer Bands of Büngner compared with WT; however, this was not statistically significant. **(G and H)** TEM micrographs of sciatic nerves from control-injected and tamoxifen-injected *Plp^Cre+;Dock1^fl/fl* mice 25 days after transection. **(I)** Quantification of the number of

remyelinated axons per 1,000 μm², **(J)** g-ratio as it relates to axon caliber between control and icKO mice, **(K)** the number of regenerated axons >1 μm per 1,000 μm², and **(L)** the number of droplets per macrophage (red arrowheads). *n* = 4 mice, 4 images per nerve (control); 4 mice, 4 images per nerve (icKO). Here, and in all figures, the X symbol in the graph denotes a data point corresponding to the representative image shown. **(A, B, G, and H)** Scale bar = 2 μm. **(C–F, I, K, and L)** Unpaired *t* test with Welch's correction. ***P < 0.001; **P < 0.01; *P < 0.05; ns, not significant.

despite having similar numbers of >1 μm regenerated axons (those large enough to be remyelinated) compared with control mice (Fig. 5 K). Lastly, we found persistent evidence of delayed debris clearance 25 dpi, with icKO mice having more lipid droplets still visible in macrophages than in controls (Fig. 5 L; control = 10.18, icKO = 23.32, P = 0.0001). Our findings in this sciatic nerve transection model complement what we observed in ZMB regeneration and further support the conclusion that Dock1 is critical for SCs to achieve timely remyelination following peripheral nerve injury.

## Rac1 inhibition enhances myelin defects in *dock1* MUTs

Rac1 is essential in SCs for radial sorting and myelination (Benninger et al., 2007; Nodari et al., 2007). Given Dock1's GEF activity for Rac1, we wanted to know whether manipulating Rac1 levels would alter myelination in *dock1*^stl45/+^ heterozygotes, which are otherwise indistinguishable from WT, or enhance the *dock1*^stl45/stl45^ homozygous MUT hypomyelination phenotype. We performed a pharmacological sensitization study using the Rac1 inhibitor EHT1864, which selectively inhibits Rac1, and to a lesser extent, Rac3 ($K_D$ 40 and 230 nM, respectively) without affecting GTPases such as RhoA and CDC42 (Onesto et al., 2008; Shutes et al., 2007). We used whole-mount in situ hybridization (WISH) for myelin basic protein (*mbp*) to assess *mbp* expression in the developing posterior lateral line (PLLn). The PLLn is a major peripheral sensory nerve that runs the length of the zebrafish and begins myelinating around 3 days postfertilization (dpf) (Sarrazin et al., 2010). We previously showed that homozygous *dock1* MUTs have a slight reduction in PLLn *mbp* expression at 5 dpf compared with WT (Cunningham et al., 2018). Consistent with our prior work, at 4 dpf, homozygous *dock1* MUTs also have slightly reduced *mbp* expression in the PLLn compared with WT, while heterozygotes are indistinguishable from WT (Fig. 6, A–C). A dose-response study (data not shown) was done by administering EHT1864 from 2 to 4 dpf; we found that at 5 μM, there was no effect on the overall health of WT zebrafish, whereas higher doses resulted in toxicity. Treating zebrafish from 2 to 4 dpf allows us to target SCs during the onset of radial sorting and the initiation of myelination. Upon examining the PLLn at 4 dpf, we saw that *mbp* expression in PLLns from WT zebrafish was unaffected by the 5 μM dose of EHT1864 (Fig. 6 D). *dock1* heterozygotes, however, showed a reduction in *mbp* expression compared with the treated WT and untreated controls (Fig. 6 E). The same was true for *dock1* MUTs, but to an even greater extent, with some segments of the PLLn completely devoid of *mbp* expression (Fig. 6 F). When we quantified our observations, we found that there were no differences in *mbp* expression between WT and *dock1* heterozygotes in our DMSO-treated controls; however, there was a correlation between the phenotypes observed and the genotypes in the EHT1864-treated zebrafish such that drug treatment enhanced *dock1* phenotypes

(Fig. 6 G). Next, we examined myelin ultrastructure by performing TEM. At 4 dpf, homozygous *dock1* MUTs had a slight reduction in the number of myelinated axons at baseline compared with WT and heterozygotes (Fig. 6, H–J). By TEM, EHT1864-treated WT zebrafish did not have a discernable change in the number of myelinated axons compared with the untreated controls (Fig. 6 K), indicating that the low drug dose we administered did not cause off-target defects. In contrast, there was a reduction in the number of myelinated axons in the *dock1* HET and homozygous MUTs following drug treatment (Fig. 6, L and M). When we quantified the total number of axons, we saw no difference between any of the genotypes before or after 5 μM EHT1864 treatment (Fig. 6 N); however, when we analyzed the number of myelinated axons, we saw that *dock1* HET and homozygous MUTs are sensitized to Rac1 inhibition (Fig. 6 O; HET control = 10.72, HET treated = 7.140, P = 0.0211; MUT control = 8.010, MUT treated = 2.889, P = 0.0014), indicating that even modest disruption to Dock1-Rac1 signaling can result in dysregulated myelination in the developing PNS.

## Rac1 interacts with Dock1 in PNS myelination

To complement our pharmacological study and further clarify the timing and specificity of potential Dock-Rac1 signaling, we employed a genetic approach using CRISPR/Cas9-mediated genome editing. Specifically, we injected high-efficiency Rac1 single guide RNA (sgRNA) and Cas9 protein into one-cell stage zebrafish embryos obtained from *dock1*^stl45/+^ intercrosses and examined F0 "crispant" animals. This is a well-established method to induce targeted mutations in genes of interest and rapidly screen phenotypes in injected animals. We performed PCR and a restriction digest to confirm that our sgRNAs were highly effective (>90% efficiency, Fig. S5 L). We examined fish using WISH and TEM at 3 dpf. In control animals, homozygous *dock1* MUTs had a slight reduction in *mbp* mRNA and the number of myelinated axons in the PLLn compared with WT and heterozygotes, which, respectively, were indistinguishable from each other (Fig. S5, A–C and K; and Fig. 7, A–C). Rac1-targeted crispant zebrafish showed a decrease in *mbp* mRNA levels and the number of myelinated axons in WT zebrafish (Fig. S5 D; and Fig. 7, D and J; control = 12.35, sgRac1 = 8.706, P = 0.0395), consistent with Rac1's role in SC myelination. Importantly, we observed a far more significant decrease in the *mbp* mRNA levels and the number of myelinated axons in the injected *dock1* HET (Fig. S5, E and K; and Fig. 7, E and J; control = 12.26, sgRac1 = 6.721, P = 0.0002) and homozygous MUTs (Fig. S5, F and K; and Fig. 7, F and J; control = 7.223, sgRac1 = 2.007, P = 0.0005) when compared with controls, with no change in the total axon number between genotypes (Fig. S5 J).

While Dock1's classically understood role is signaling to Rac1, there is evidence that Rac3, another member of the Rac subfamily, can also be regulated by GEFs similar to Dock1 (Marei

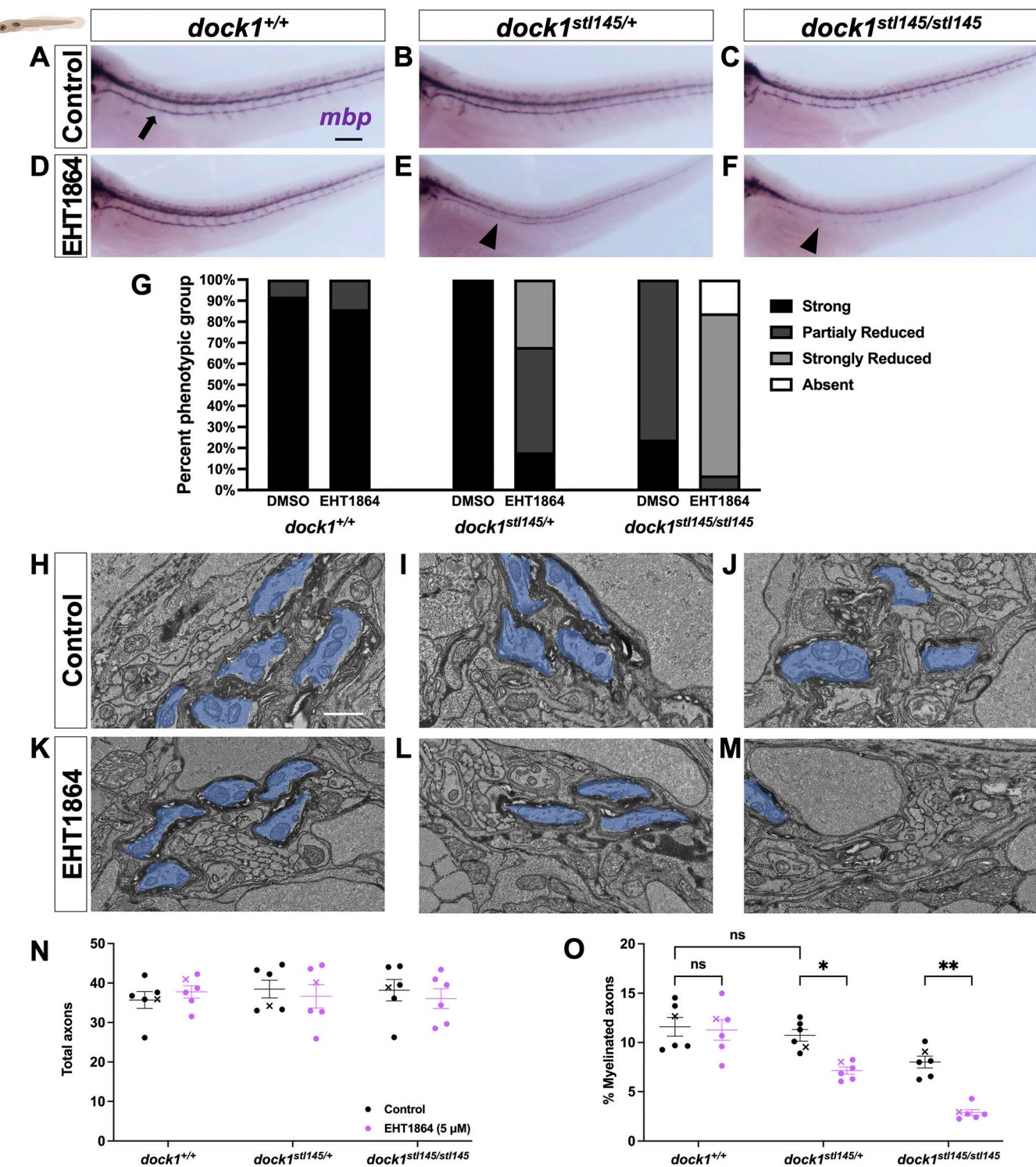

Figure 6. **dock1 MUT zebrafish are sensitized to Rac1 inhibition. (A–F)** Lateral views of larvae showing *mbp* expression by WISH in DMSO control-treated and EHT1864-treated WT *dock1+/+*, HET *dock1stl145/+*, and homozygous *dock1stl145/stl145* MUT zebrafish. **(A)** The arrow points to strong *mbp* expression in the PLLn. **(E and F)** Arrowheads highlight decreased mbp expression in the PLLn. **(G)** The quantification of WISH was assessed by examining *mbp* expression along the entire PLLn in 4 dpf DMSO and EHT1864 zebrafish, comparing phenotypic scores and genotypes. **(H–M)** TEM micrographs of cross-sections of the PLLn, showing myelinated axons pseudocolored in blue, in DMSO control-treated and EHT1864-treated WT *dock1+/+*, HET *dock1stl145/+*, and homozygous *dock1stl145/stl145* MUT zebrafish. **(N and O)** Quantifications of the total axons and myelinated axons in the PLLn. *n* = 6 fish per genotype (DMSO control) and 6 fish per genotype (EHT1864). Here, and in all figures, the X symbol in the graph denotes a data point corresponding to the representative image shown. **(A–F)** Scale bar = 100 μm. **(H–M)** Scale bar = 1 μm. **(N and O)** Two-way ANOVA with Sidak's multiple comparisons test. **P < 0.01; *P < 0.05; ns, not significant.

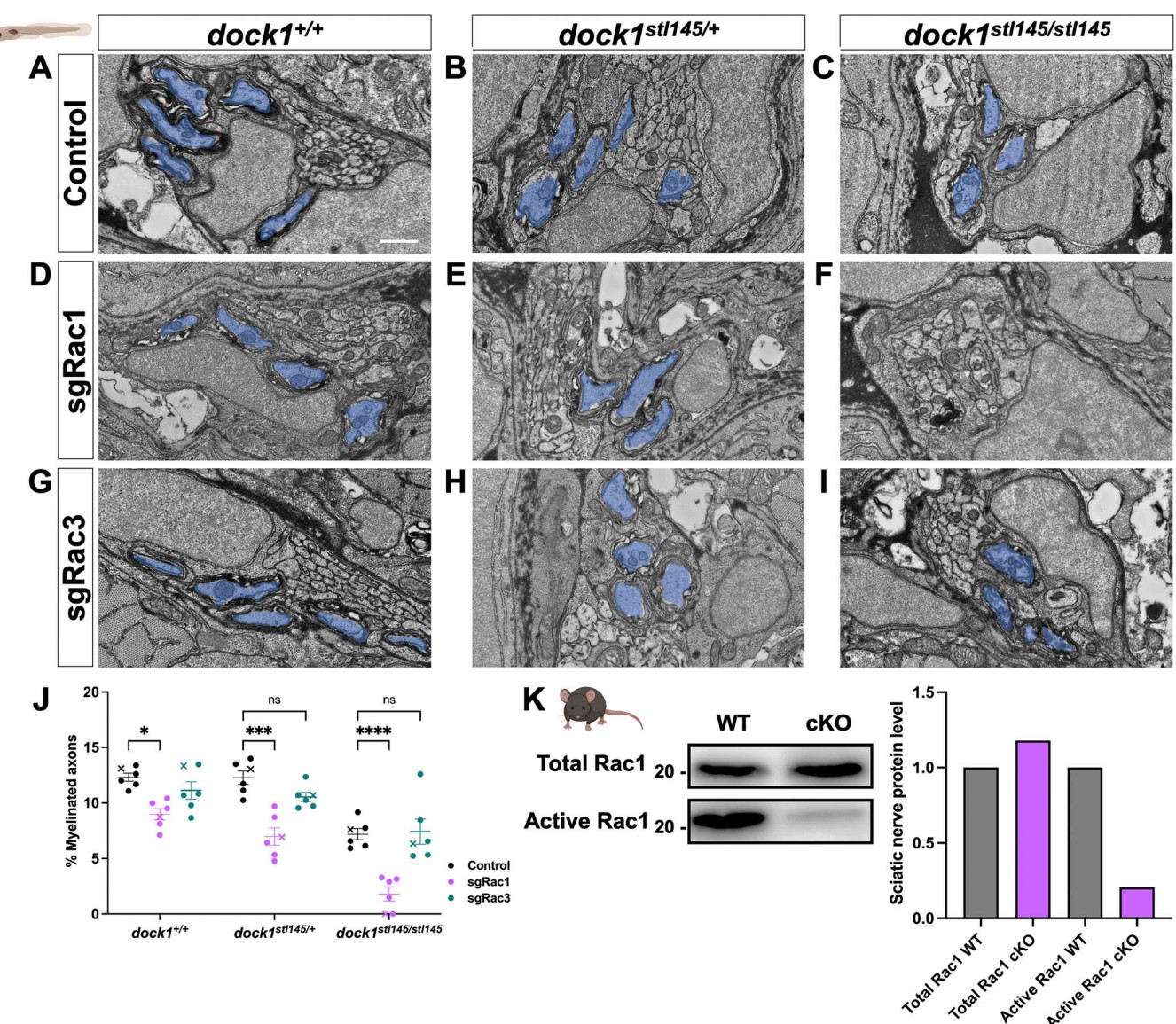

Figure 7. **Targeted genetic approaches reveal an interaction between Dock1 and Rac1 in the developing PNS. (A–I)** TEM micrographs of cross-sections of the PLLn, showing myelinated axons pseudocolored in blue, in control (Cas9 only), sgRac1+Cas9, and sgRac3+Cas9 in WT *dock1⁺/⁺*, HET *dock1^stl145/+*, and homozygous *dock1^stl145/stl145* MUT zebrafish. **(J)** Quantification of myelinated axons in the PLLn. *n* = 6 fish per genotype and experimental condition. **(K)** Western blot showing sciatic nerve active and total Rac1 protein levels from control and *Dock1* cKO animals, along with quantification of protein levels. Representative blot consisting of 12 nerves from 6 animals per genotype. Here, and in all figures, the X symbol in the graph denotes a data point corresponding to the representative image shown. **(A–I)** Scale bar = 100 μm. **(J)** Two-way ANOVA with Sidak's multiple comparisons test. ***P < 0.001; *P < 0.05; ****P < 0.0001; ns, not significant. Source data are available for this figure: SourceData F7.

and Malliri, 2017). A recent study found that deleting Rac3 partially rescues the axonal sorting and myelination defects caused by Rac1 loss; however, these beneficial effects decline with age (Pellegatta et al., 2021). While the $K_D$ of EHT1864 for Rac3 is over 5× higher than for Rac1 (230 versus 40 nM), indicating significantly lower binding of Rac3, we still wanted to rule out the possibility of compound Rac1+Rac3 inhibition in the pharmacogenetic experiments. To test this, we genetically inhibited *rac3* by injecting highly efficient Rac3 sgRNA (Fig. S5 M) and Cas9 protein into 1-cell embryos obtained from *dock1^stl45/+* intercrosses. We saw no significant differences in the *mbp* RNA levels, the number of myelinated axons, or the total number of

axons when we compared Rac3 sgRNA + Cas9-injected WT versus *dock1* HET and *dock1* homozygous fish (Fig. S5, G–K and Fig. 7, G–J).

Finally, to determine if *Dock1* MUT SCs have reduced levels of active Rac1, we performed a pulldown of sciatic nerve lysate from P3 mice and compared the amount of active, GTP-bound Rac1 from cKO MUT and control mice. We found that active Rac1 levels in *Dock1* cKO sciatic nerves were strongly decreased relative to WT controls (Fig. 7 K). Taken together, our data support the model that Dock1-Rac1 signaling in SCs is crucial for proper developmental myelination.

## Discussion

We previously established Dock1 as an important regulator of developmental myelination in zebrafish and showed that a global mutation in *dock1* results in early developmental hypomyelination (Cunningham et al., 2018). In the present study, we used zebrafish and mouse models to more fully define the role of Dock1 in PNS myelination. In zebrafish, we analyzed adult animals to look beyond development, where we found that Dock1 is vital for the long-term maintenance of myelin health and in repair and remyelination following nerve injury. In a complementary series of experiments in mice, we found that the observations made in global *dock1* zebrafish MUTs stem from an evolutionarily conserved function of Dock1, where we showed that it acts cell autonomously in murine SCs.

### A unique model to study myelin in the adult zebrafish PNS

To study myelin in adult zebrafish, we turned to a system that has been characterized but has yet to be used as an experimental tool, the ZMB. We found that the myelin of *dock1*$^{stl45/stl45}$ homozygous MUTs was indistinguishable from WT in early adulthood but that these MUTs had accumulated a significant number of myelin abnormalities at 12 mo of age. Next, we used the regenerative capabilities of the ZMB to assess if Dock1 functions in remyelination following nerve injury. We found that homozygous MUTs regenerated the same total number of axons; however, there was a significant reduction in remyelinated axons 28 days after transection. These findings expanded our understanding of Dock1's importance in the zebrafish PNS; however, we could not assign specific functions Dock1 might have in a particular cell type since the experiments were performed in global MUTs.

### Dock1 functions cell autonomously in SCs to regulate PNS myelination

To determine whether Dock1 functions cell autonomously in SCs and if our findings in zebrafish were evolutionarily conserved in mammals, we generated SC-specific *Dock1* knockout mice by crossing two validated and well-characterized mouse lines (Jaegle et al., 2003; Laurin et al., 2013; Leone et al., 2003). SC-specific *Dock1* knockout mice had reductions in myelin thickness at P3 during early development and abnormal SC morphology. SC-specific *Rac1* MUT mice have similar phenotypes as SC-specific *Dock1* MUTs, including signs of delayed radial sorting and early developmental hypomyelination (Benninger et al., 2007; Guo et al., 2012; Nodari et al., 2007). Additionally, *Dock1* MUT SCs phenocopy *Rac1* (Benninger et al., 2007; Guo et al., 2012; Nodari et al., 2007) and *Gpr126/Adgrg6* (Mogha et al., 2013) MUT SCs in terms of aberrant cytoplasmic protrusions and accompanying basal lamina trails. When we examined MUT mice at P28, we found the hypomyelination was no longer present; however, when we looked at 12 mo, we saw a significant increase in myelin abnormalities, similar to what we observed in fish. It is not uncommon for genes important in development to also play a role in myelin maintenance (Bremer et al., 2011; Decker et al., 2006). The fact that we observe early and late phenotypes in *Dock1* MUTs could suggest that the cytoskeletal abnormalities that give rise to developmental myelin defects

resolve in early adulthood, perhaps due to compensation by other Dock family members (more on this below), but become dysregulated again in mature animals. Myelin abnormalities seen in *Dock1* MUTs in both zebrafish and mice associated with aging could occur due to impaired downstream Rac1 signaling, which is important in maintaining SC plasticity or adaptability in mature animals (Xu et al., 2021). Rac1 also has roles in reactive oxygen species production, which can lead to oxidative stress if not properly regulated (Bailly et al., 2024). With aging, SCs are more prone to oxidative stress and damage (Fuentes-Flores et al., 2023; Painter et al., 2014). Disrupted Rac1 signaling could increase susceptibility to oxidative stress in the sciatic nerve, damaging myelin and contributing to the age-related deterioration of myelin integrity over time.

RhoGTPases, including Rac1, regulate many of the signaling pathways in SCs associated with repair and remyelination, including MAP kinases and c-Jun (Harrisingh et al., 2004; Park and Feltri, 2011; Syed et al., 2010). To assess the function of Dock1 in mammalian PNS repair, we performed sciatic nerve transections, a method often used to examine debris clearance and remyelination, which more closely aligns with the ZMB transection model than a nerve crush injury. In *Drosophila*, the ortholog of Dock1, known as CED-5, operates in conjunction with CED-2 and CED-12, homologs of mammalian CrkII and Elmo, respectively. This complex functions as a GEF to activate downstream Rac1 (Ziegenfuss et al., 2012). Disruption of CED-2/CED-12 signaling, or a parallel pathway, leads to suppression of the engulfment and degradation of cellular debris. Along these lines, when we performed sciatic nerve transection in mice, *Dock1* icKO mice had more foamy macrophages, more macrophages that still contained intact myelin cylinders, and more motile appearing macrophages than controls, all signs associated with delayed debris clearance following nerve injury (Arthur-Farraj et al., 2012; Fry et al., 2007). When we looked later to assess remyelination, we found that *Dock1* icKO mice exhibited a significant decrease in remyelinated axons 25 dpi compared with controls, similar to zebrafish ZMB studies, as well as persistent signs of delayed debris clearance, thus demonstrating a crucial and evolutionarily conserved role for Dock1 in SCs during repair and remyelination.

### What are the signaling partners of Dock1?

The Rho-GTPase Rac1 has been extensively characterized for its role in modulating cellular morphological transformations, primarily by orchestrating cytoskeletal dynamics through actin polymerization. This function has implications in SC development, where differential Rac1 expression regulates the timing of SC migration, radial sorting, and myelination (Benninger et al., 2007; Guo et al., 2012; Nodari et al., 2007). Dock1 is known to exert GEF activity on Rac1 (Côté and Vuori, 2002); however, the relationship between Dock1 and Rac1 signaling has yet to be examined in the context of myelination. Returning to our zebrafish models, we asked whether *dock1*$^{stl45/+}$ heterozygotes, whose *mbp* expression and myelin morphology are indistinguishable from WT, would be sensitized to Rac1 inhibition. This was precisely the case, with low-level Rac1 inhibition leading to a reduction of *mbp* expression, the number of myelinated axons

in *dock1*<sup>stll45/+</sup> heterozygotes, and an enhancement of the *dock1*<sup>stll45/stll45</sup> homozygous MUT phenotypes. In addition, our experiments using a targeted genetic approach to manipulate Rac1 in zebrafish and our analysis of active Rac1 levels in developing mice provide additional support for a model where Dock1 is an interacting partner of Rac1 in PNS myelination.

It is established that Dock1 binds to the adapter protein Elmo1, an interaction that stabilizes the connection with Rac1 and directs the assembled protein complex to the plasma membrane, where it regulates the cytoskeleton (Brugnera et al., 2002; Grimsley et al., 2004; Komander et al., 2008; Lu et al., 2004; Lu and Ravichandran, 2006; Mikdache et al., 2020). Despite the fundamental role of Rac1 in SCs, the precise subcellular site of its activation has yet to be determined. The radial sorting abnormalities in *Rac1* MUTs are shared with MUTs that influence proteins tied to the basal lamina of the SC, like those found in laminin MUTs (Chen and Strickland, 2003). This observation may suggest a potential abaxonal positioning for Rac1. Conversely, since the SC's plasma membrane extensions during radial sorting demand the intertwining of processes into axonal bundles, one might also infer that the localization of the active Rac1 signal could be on the adaxonal side, where the SC directly interacts with the axon. Understanding the specifics of this signaling will provide valuable insight into SC development and enhance our understanding of how SCs sort axons.

Since myelination during development is important for the normal function of the PNS, having multiple GEFs regulate this process and intersect at the same pathway could provide built-in redundancy and a biological advantage, permitting radial sorting and forming myelin even if a single GEF functions abnormally. This may help explain why the developmental phenotypes we observed in zebrafish and mice resolve in early adulthood. Dock1 might operate with other GEFs, which could be upregulated or act redundantly when it is nonfunctional to ultimately control Rac1 levels. This redundancy may come from other members of the Dock1 family, such as Dock7 or Dock8, which have been shown to have roles in regulating SC migration and development (Miyamoto et al., 2016; Yamauchi et al., 2008, 2011).

Proteins upstream of Dock1 are not well defined. It is known, however, that RhoGEFs can be activated by and function downstream of receptor tyrosine kinases (RTKs), and accordingly, Dock1 has been suggested to function downstream of RTKs in several biological contexts (Duchek et al., 2001; Feng et al., 2012). For example, ErbB2 (HER2) interacts with DOCK1 in breast cancer cells (Laurin et al., 2013). In the context of SCs, the ErbB2/3 heterodimer is the most thoroughly investigated RTK pair, exhibiting critical functions across multiple developmental phases, encompassing migration, radial sorting, and myelination (Monk et al., 2015). The developmental stages regulated by ErbB2/3 in SCs require dramatic cell shape changes and process extension, and as previously noted, Dock1 regulates similar cell shape changes in many biological systems. Additionally, ErbB2, through Rac1 and Cdc42, has been shown in vitro to activate Dock7 to regulate SC migration (Yamauchi et al., 2008), positioning ErbB2/3 as a promising candidate for an upstream signaling partner of Dock1. Alternatively, Dock1 functions downstream of chemokine G protein–coupled receptor (GPCR)

signaling in endothelial cell migration (Sanematsu et al., 2010), and the Dock1 adapter protein Elmo1, which has been shown to regulate zebrafish PNS myelination (Mikdache et al., 2020), directly interacts with adhesion GPCRs Bai1/Adgrb1 and Bai3/Adgrg3 in myoblast fusion (Hamoud et al., 2014; Hochreiter-Hufford et al., 2013). Interestingly, SC-specific *Dock1* MUT mice phenocopy the abnormal cytoplasmic protrusions observed in SCs that lack Gpr126/Adgrg6, another adhesion GPCR that is essential for SC myelination (Mogha et al., 2013). In the future, it will be interesting to assess if Gpr126/Adgrg6, which is required for timely radial sorting and essential for SC myelination (Monk et al., 2009), is an upstream activator of Dock1.

In summary, our work combines a series of in vivo experimental approaches from zebrafish and mice to demonstrate that Dock1 plays an evolutionarily conserved, cell-autonomous function in SCs and interacts with Rac1 to regulate PNS myelin biology. When Dock1 is not functional in SCs, myelination is dysregulated during development, myelin abnormalities arise in late adulthood, and SCs lose their ability to repair and remyelinate the PNS after nerve injury. These findings provide crucial insights into our understanding of SC and PNS myelin and offer valuable directions for future studies, which will ultimately help us develop better therapeutic interventions.

## Materials and methods
### Zebrafish lines and rearing conditions
All animal experiments and procedures in this manuscript were performed in compliance with the institutional ethical regulations for animal testing and research at Oregon Health and Science University. *dock1* transgenic zebrafish (Cunningham et al., 2018) are maintained as heterozygotes (*dock1*<sup>stll45/+</sup>), an incross of which yields WT, HET, and homozygous viable zebrafish. Zebrafish larvae are fed a diet of rotifers and dry food (Gemma 75) from 5 dpf until 21 dpf. From 21 dpf until 3 mo, fish are fed using rotifers and dry food (Gemma 150). Adult fish are maintained and fed with brine shrimp and dry food (Gemma 300). For larval zebrafish studies, sex cannot be considered as a biological variable as sex has not yet been determined in this species. For experiments using adult zebrafish, equal numbers of males and females were examined.

### Mouse strains and maintenance
All mice used—*Dock1*<sup>fl/fl</sup> (Laurin et al., 2013), *Dhh*<sup>Cre</sup> (Jaegle et al., 2003), and *PLP*<sup>Cre-ERT2</sup> (Leone et al., 2003)—are previously described and validated lines. For experiments using *Dhh*<sup>Cre</sup> (cKO), *Dhh*<sup>Cre+</sup>;*Dock1*<sup>fl/+</sup> mice were crossed to Dock1<sup>fl/fl</sup> mice to generate *Dhh*<sup>Cre+</sup>;*Dock1*<sup>fl/fl</sup> mice and their sibling controls. For experiments using *Plp*<sup>Cre-ERT2+</sup> (icKO), *Plp*<sup>Cre-ERT2+</sup>;*Dock1*<sup>fl/+</sup> mice were crossed to Dock1<sup>fl/fl</sup> mice to generate *Plp*<sup>Cre-ERT2+</sup>;*Dock1*<sup>fl/fl</sup> mice and their sibling controls. To induce Cre recombination and Dock1 deletion in mice, 2-mo-old *Plp*<sup>Cre-ERT2+</sup>;*Dock1*<sup>fl/fl</sup> mice were injected for five consecutive days with corn oil (control) or 100 mg/kg of tamoxifen (T5648; Sigma-Aldrich) dissolved in corn oil (icKO). For all mouse experiments, mice of both sexes were analyzed, and MUTs were always compared with littermate sibling controls.

## Genotyping

Zebrafish—*stl145* primers were used to amplify a mutation-spanning region by PCR: F: 5′-CATAGGCGTTCTTCACTGAG-3′ and R: 5′-GACAACAGCTGCCTAATCCG-3′. After PCR, a restriction enzyme digest assay was performed, and the resulting fragments were analyzed on a 3% agarose gel. The *stl145* C-to-T mutation disrupts a BstNI site so that the WT PCR product is cleaved into 48- and 353-bp products, and the MUT PCR product is uncut at 401 bp. Mice—The following primers to detect the presence of the alleles: *Dock1^fl/fl*, 5′-TCAGCAGGCCCAGTTCCTACT-3′; 5′-GCAGAGCTAGGAGTTCATCGTAGTTC-3′, *Dhh^Cre*, 5′-CCTTCTCTATCTGCGGTGCT-3′; 5′-ACGGACAGAAGCATTTTCCA-3′, and *PLP^Cre-ERT2*, 5′-CACTCTGTGCTTGGTAACATGG-3′; 5′-TCGGATCCGCCGCATAAC-3′. After PCR, the resulting products were analyzed on a 3% agarose gel.

## ZMB transection

Adult zebrafish were anesthetized with 0.16 mg/ml tricaine diluted in system water, placed onto a SYLGARD (Dow Chemical)-filled plate, and visualized under a stereomicroscope (Zeiss Stemi 508). A pair of fine forceps was used to grab the distalmost tip of the barbel and lift it away from the surface of the fish. The cut was performed by placing a pair of microdissection scissors parallel to the mouth's surface to ensure consistency in the cut site between animals. Once removed, the barbel was placed into modified Karnovsky's fix (2% glutaraldehyde and 4% PFA in 0.1 M sodium cacodylate, pH 7.4), kept on ice, and processed as described below. Fish were returned to individually housed tanks to track them during the regeneration period, and the same procedure was repeated 28 days later, this time removing maxillary barbels from each side.

## Sciatic nerve transection

Mice were deeply anesthetized with isoflurane before and during surgery. The fur on the right hind limb and lower back was removed with an electric razor, and the sciatic nerve of the right hind limb was exposed by making a small cut in the skin on the upper thigh. This initial cut in the skin exposed a thin layer of muscle through which the location of the sciatic nerve could be seen, and a second small cut was made in the muscle to allow access to the sciatic nerve. Effort was made to minimize the size of the incision and the disruption to the surrounding muscle during surgery. The exposed sciatic nerve was quickly crushed ~500 μm distal to the sciatic notch with forceps coated in powdered carbon to mark the injury site and then transected using a pair of fine microdissection scissors at that location. The nerve retracted in each direction following transection due to its elasticity, and while the retraction can be variable, it was within 200–400 μm for all samples. In all cases, the nerve endings remained oriented toward each other in the sciatic nerve tract. The transection site was sealed with synthetic absorbable sutures placed in the muscle, followed by nylon sutures and metal clips to close the incision in the skin. Mice were monitored daily and administered pain-reducing chow (Bio-Serv) during recovery until they were euthanized for experimental endpoints.

## TEM

Zebrafish—Zebrafish larvae and adult barbels were processed as follows. For larvae, zebrafish were anesthetized with tricaine and cut between body segments 5 and 6 to control for variability along the anterior-posterior axis. For ZMBs, the structures were removed by placing a pair of microdissection scissors parallel to the skin to ensure a consistent cut as close to the facial surface as possible. Samples were immersed in a modified Karnovsky's fix (2% glutaraldehyde and 4% PFA in 0.1 M sodium cacodylate, pH 7.4) and microwaved (PELCO BioWave processing—Ted Pella) at 100 W for 1 min, OFF for 1 min, 100 W for 1 min, and OFF for 1 min, 450 W for 20 s, and OFF for 20 s. This was repeated five times, and samples were allowed to fix overnight at 4°C. The following day, samples were rinsed three times in 0.1 M sodium cacodylate buffer at RT, 10 min each rinse. A secondary fixative solution of 2% osmium tetroxide was prepared by combining 2 ml of a stock 0.2 M sodium cacodylate + 0.2 M imidazole solution (pH 7.5) with 2 ml of 4% osmium tetroxide. The 2% osmium tetroxide was added to the samples, and they were microwaved –100 W for 1 min, OFF for 1 min, 100 W for 1 min, OFF for 1 min, 450 W for 20 s, and OFF for 20 s. This was repeated five times, and we allowed them to sit for an additional 2 h at RT. The osmium tetroxide was removed, and the samples were washed three times with deionized water, 10 min per wash. UranyLess (Electron Microscopy Sciences) was then added to the tubes, and the microwave was run –450 W for 1 min, OFF for 1 min, and 450 W for 1 min. The samples remained in UranyLess overnight at 4°C. The following day, the UranyLess was removed, and the samples were washed three times with deionized water, 10 min per wash. A series of ethanol:water (25:75, 50:50, 70:30, 80:20, 95:5, and 100:0) solutions were prepared. Samples were then passed through this graded series of increasing ethanol concentrations, 25% EtOH, 50% EtOH, 70% EtOH, 80% EtOH, and 95% EtOH, and were microwaved –250 W for 45 s followed by incubation at RT for 10 min for each concentration. Next, they were changed into a 100% EtOH solution and microwaved at 250 W for 1 min, OFF for 1 min, and then 250 W for 1 min; then incubated at RT for 10 min. This step was repeated with the 100% EtOH two more times, for 3× in the 100% EtOH. Next, samples were dehydrated using 100% EM grade acetone and microwaved –250 W for 1 min, OFF for 1 min, and 250 W for 1 min and incubated at RT for 10 min. This step was repeated with the 100% acetone two more times, for 3× in the 100% acetone. Next, a 1:1 solution of Araldite 812:100% acetone was added to the samples and allowed to infiltrate at RT overnight. The following day, a fresh batch of Araldite 812 was prepared. With the aid of a stereomicroscope (Zeiss Stemi 508), the samples were carefully oriented in molds so that they were properly aligned for sectioning. They sat at RT for 4–6 h in the Araldite 812 before being placed in a 65°C oven and allowed to polymerize for a minimum of 48 h.

Mice—Sciatic nerves were removed from mice and fixed in modified Karnovsky's fix (2% glutaraldehyde and 4% PFA in 0.1 M sodium cacodylate, pH 7.4) at 4°C overnight. Nerves were pinned down in a SYLGARD-filled dish using 0.20-mm insect pins (Austerlitz) to ensure that they fixed straight. 4-0 nylon sutures were tied around the distal end of the nerve and

removed at the time of embedding to ensure correct cutting orientation. Following fixation, nerves were rinsed three times, 15 min each, in 0.1 M sodium cacodylate buffer and then post-fixed with 2% osmium tetroxide (as described above) overnight at 4°C. Nerves were then dehydrated in a graded ethanol series (25%, 50%, 70%, 95%, and 100%) 3× for 20 min per solution. An additional 20 min 50:50 ethanol:propylene oxide and 2 × 20 min 100% propylene oxide dehydrations were performed before overnight incubation in 50:50 Araldite 812:propylene oxide mixture. This was followed by transitions to a 70:30 and a 90:10 Araldite 812:propylene oxide mix; each kept overnight at 4°C before proceeding to the next.

On the final day, nerves were put in 100% Araldite 812, allowed to sit at RT for several hours to allow infiltration, placed in labeled molds, and baked for a minimum of 48 h at 65°C. For all zebrafish and mouse samples, semithin sections (400 nm) were stained with toluidine blue and viewed on a light microscope (Leica DM 300) to ensure quality before cutting for TEM. Ultrathin sections (60 nm) were cut and placed on 100 mesh Formvar grids (#FCF100-Cu; EMS) and counterstained with UranyLess (Electron Microscopy Sciences) and 3% Lead Citrate (Electron Microscopy Sciences), and then images were acquired on an FEI Tecnai T12 TEM microscope using an Advanced Microscopy Techniques CCD camera.

### WISH

Zebrafish were fixed in 4% PFA (made in 1X PBS) at RT for 2 h with shaking. The PFA was replaced with 100% MeOH, 5 × 5 min each in 100% MeOH. After the final wash, embryos were stored in 100% MeOH at –20°C until they were ready to be processed. On day 1 of processing, embryos were rehydrated into PBSTw (PBS + 0.1% Tween 20) (50% PBSTw—70% PBSTw—100% PBSTw) with 5 min washes each, followed by 4 × 5 min PBSTw washes at RT with shaking. Samples were placed in 1:2,000 ProtK liquid stock (20 mg/ml) in PBS without shaking for 55 min. The ProtK was removed, followed by 2 × 5 min PBSTw washes to remove the ProtK. Samples were postfixed in 4% PFA for 20 min at RT with shaking. The PFA was removed, and there were 5 × 5 min washes in PBSTw at RT with shaking. Next, the samples were prehybridized in 400 µl Hyb(+) solution for 1–2 h at 65°C. Tubes were kept on their sides to ensure adequate exposure to the solution. Next, 400 µl of probe diluted in Hyb(+) was added to each tube and left overnight at 65°C, with the tubes on their sides. On day 2, all the solutions used were preheated to 65°C before adding them to the samples, and all washes were done at 65°C, taking care to ensure the samples were not allowed to cool down. The probe was removed, 100% Hyb was added, and the samples were left to sit for 5–10 min. Next, a series of liquid changes were performed: 75% Hyb:25% 2X SSCTw 5 min at 65°C, 50% Hyb:50% 2X SSCTw 5 min at 65°C, 25% Hyb:75% 2X SSCTw 5 min at 65°C, 2X SSCTw 2 × 30 min at 65°C, 0.2X SSCTw 2 × 30 min at 65°C, and Maleic Acid Buffer + 0.1% Triton (MABTr) 10 min at RT, shaking tubes on side. A blocking solution was prepared by combining 2% blocking reagent in MABTr +10% sheep serum. The block was added to the samples and incubated for 1–2 h at RT while shaking tubes on their side. Next, the block was removed and replaced by anti-Dig AP Fab fragments, diluted 1:2,000 in blocking solution and left overnight at 4°C with shaking. On day 3, the anti-Dig AP Fab fragment solution was removed, and MABTr washes were performed: 6 × 30 min each at RT with shaking. The MABTr was removed, and alkaline phosphatase/NaCl, tris-HCL, MgCl$_2$, tween 20 (AP/NTMT) buffer was added and allowed to sit for 10 min at RT with shaking. The samples were then incubated in a solution of AP buffer + 4-nitro blue tetrazolium chloride (NBT) (2.2 µl/ml AP buffer) + 5-bromo-4-chloro-3-indolyl-phosphate (BCIP) (1.6 µl/ml AP buffer) and covered in foil as the reaction is light sensitive. The reaction proceeded for 2–3 h until the signal along the lateral line was visible under a light microscope, and the reaction was stopped by doing three quick washes in PBSTw. The samples were then postfixed in 4% PFA for 30 min at RT. The PFA was removed, and samples were passed through a 30%–50%–70% glycerol series, moving on to the next one after they sank to the bottom. Samples were stored in 70% glycerol at 4°C until they were ready to be imaged. Samples were mounted onto slides, suspended in 70% glycerol, and brightfield imaged using a Zeiss Discovery.V8 stereomicroscope.

### EHT1864 treatment

Zebrafish larvae were treated with 0.003% phenylthiourea at 24 hours postfertilization (hpf) to inhibit pigmentation. At 48 hpf, immediately before treatment, zebrafish were manually dechorionated with a pair of #5 forceps (Fine Science Tools). Zebrafish were treated with DMSO or 5 µM EHT1864 (3872; Tocris) in embryo media with 0.003% phenylthiourea from 48 to 98 hpf. At 96 hpf, larvae were anesthetized with tricaine and processed for WISH or TEM as described.

### Western blotting

Mice were euthanized, and sciatic nerves from both legs were harvested, placed together in labeled 1.7-ml tubes, and immediately flash frozen on dry ice and stored at –80°C until the protein extraction occurred. The sciatic nerves were thawed, and a solution of radioimmunoprecipitation assay (RIPA) buffer (50 mM Tris HCl, pH 8.0, 150 mM NaCl, 1% NP-40, 0.5% sodium deoxycholate, 0.1% SDS, 1 mM EDTA, and 0.5 mM EGTA) containing protease inhibitor (11836153001; Roche) was added to the nerves. A sterile tissue homogenizer was attached to a Ryobi drill press, and the nerves were homogenized by moving the homogenizer up and down in the dounce 30 times until the nerve appeared completely homogenized. The dounce was kept in a beaker of ice water during this process, and care was taken to ensure that the tissue remained cold throughout homogenization. The samples were allowed to rest on ice for 10 min, then spun for 15 min at 15,000 rpm in a 4°C cooled centrifuge. The supernatant was then removed and added to a new tube. A Bradford protein assay was performed to ensure equal protein concentrations in our samples before proceeding. BCA standards are combined with Milli-Q water and 1 ml of Coomassie Plus. The standards used were 750, 500, 250, 125, 65, and 0 µg/ml. For each tube, 5 µl of sample and 495 µl of Milli-Q water were added to a cuvette, along with 1 ml of Coomassie Plus. Samples were measured on a Nanodrop spectrophotometer following the measurement of a blank sample. The spread between the lowest

and highest protein concentrations was <5%. Next, 1 part Laemmli buffer was combined with four parts of protein w/RIPA buffer, and samples were thoroughly mixed. Samples were heated for 5 min to 95°C and briefly spun. The gel tank was assembled, and a 4–12% Bis-Tris Gel (NP0335BOX; Invitrogen) was loaded with ladder 9 and 25 µl of sample per lane. The gel was run at 150 V for 1 h. The gel was then removed and placed into a sandwich with a polyvinylidene fluoride (PVDF) membrane (IPVH00010; Thermo Fisher Scientific) and sponge pads in a gel blotting cassette (A25977; Thermo Fisher Scientific). The transfer was run at 20 V for 1 h. The membrane was placed in a black box and washed with 1× TBS with 0.1% Tween-20 (TBST) for 10 min. The membrane was then blocked with 5% milk powder in 1× TBST on a shaker at RT for 1 h. The membrane was then transferred into Dock1 primary antibody (1:1,000, 23421-1-AP; Proteintech) made in 1× TBST with 2% BSA and incubated overnight at 4°C with shaking. The following day, the primary antibody was removed. The membrane was washed 3×, 5 min each, with TBST, and following the final wash, HRP conjugate goat anti-rabbit secondary (7074; Cell Signaling), 1:2,000 in 1× TBST with 2% milk powder, was added. The membrane was incubated at RT for 2 h and rinsed 3× with TBST and 1× with TBS. The membrane was visualized using a chemiluminescence reaction (34080; Thermo Fisher Scientific) and imaged with a Syngene GBox iChemiXT. Following imaging, membranes were washed with TBST and re-probed with HRP-conjugated β-actin (A3854; Sigma-Aldrich). Densitometric analysis was performed in Fiji by quantifying the intensity of the Dock1 protein bands relative to the β-actin loading control and then normalized relative to the controls.

## Genetic disruption of Rac1 and Rac3

The sgRNA sequences targeting *rac1a* exon 2 (5′-GAGAGGTCA GCTTACACAGTGGG-3′) and *rac3a* exon 1 (5′-AGTGTGTTGTTG TCGGCGACGGG-3′) were selected using CHOPCHOP, based on their high predicted cutting efficiency. We injected embryos obtained from an incross of *dock1stl145/+* animals at the single-cell stage with either *rac1a* or *rac3a* sgRNA + Cas9 protein to disrupt the corresponding gene product. Control animals were injected with Cas9 protein alone. Guide efficiency was assessed by PCR amplification and restriction digest. For *rac1a*, the following primers were used: 5′-ATCAAACTCCAGTCACTGCAAA-3′ and 5′-CATGCCTTCTGATCAGCTACAC-3′, producing a 286-bp amplicon. *rac1a* disruption was confirmed by failed cutting due to loss of the BspCNI restriction site. For *rac3a*, we used primers 5′-GGGTACCTCCACATCTCATTTC-3′ and 5′-TAATACGGG AAACGACACAGAA-3′ to generate a 289-bp product, with disruption validated by the loss of the BccI restriction site. PCR products and restriction digests were analyzed on an agarose gel to confirm efficiency. Zebrafish embryos were analyzed at 3 dpf by WISH or transmission TEM as described above, genotyped, and confirmed to have sgRNA-induced mutations.

## Active Rac1 pull-down and activity assay

The left and right sciatic nerves from 6 *Dhh^Cre+;Dock1^fl/fl* and 6 *Dhh^Cre+;Dock1^+/+* littermate controls (12 nerves each genotype) were harvested and flash frozen on dry ice. Active Rac1 was measured using the Thermo Fisher Scientific Active Rac1 Pull-Down and Detection Kit (16118; Thermo Fisher Scientific). The nerves were combined in a 1.7-ml tube with 800 µl of the provided lysis/binding/wash buffer and homogenized for 3 min using a pellet mixer (47747-370; VWR). The tube was vortexed and spun at 15,000 g at 4°C for 15 min, and the supernatant was transferred to a new tube. The pulldown was performed as per the manufacturer's instructions, and the protein was analyzed by western blot, performed as described above. The experiment was performed twice and results from one technical replicate are shown. Densitometric analysis was performed in Fiji by quantifying the intensity of the active GTP-bound Rac1 protein bands relative to the total Rac1.

## Morphological characterizations

For determining the percent of axons with abnormal myelin in zebrafish at 4 and 12 mo of age, the number of myelinated axons with disrupted myelin sheaths (splitting and degeneration) was divided by the total number of myelinated axons. For TEM analysis in mice: to calculate g-ratios, we manually measured axon diameter and axon-plus-myelin diameter in ImageJ. We measured a minimum of 100 axons from 3 ~2,000 µm² regions of each sciatic nerve selected at random. The measurements were taken with the observer blind to treatment. To determine the % of axons with abnormal profiles in mice, the number of abnormally myelinated axons (outfolding, degeneration, and decompaction) was divided by the total number of myelinated axons. In addition, the percentage of abnormal Remak bundles and the number of degenerating axons compared with controls were included. In quantifying sciatic nerve injury phenotypes, we define intact myelin cylinders as identifiable uninterrupted myelin rings within a macrophage. For quantification of in situ hybridization, larval zebrafish were imaged and assigned values of "strong," "partially reduced," "strongly reduced," and "none" based on *mbp* expression along the entirety of the lateral line (Cunningham et al., 2018).

## Statistical analysis

Quantifications and assessments for all experiments were performed blinded to genotype and treatment conditions. All statistical analyses were performed using GraphPad Prism 10. For zebrafish barbel morphometric analysis, cross sections of the entire barbel were analyzed. For an analysis of the effect of two variables (genotype and age) or (genotype and control/injured), two-way ANOVA was used with Tukey's or Sidak's multiple comparisons test to analyze the effect of genotype and experimental condition compared with controls. When comparing multiple experimental groups to the same control group, a one-way ANOVA with a Brown–Forsythe test was used. When comparing one experimental group to a control, we used an unpaired *t* test with a Welch's correction. For quantifying *mbp* expression by WISH, an average for each score per genotype and condition was calculated, and a Chi-squared analysis was performed to determine significance. P values shown are represented as follows: *P < 0.05; **P < 0.01; ***P < 0.001; ****P < 0.0001; ns, not significant. Parametric data distribution was assumed to be normal, but it was not formally tested. The "X" symbol in graphs denotes a data point corresponding to the representative image shown.

## Online supplemental material

Fig. S1 shows that the *dock1* MUT zebrafish exhibit no myelin defects at 4 mo of age. Fig. S2 shows that the regenerated barbels of adult *dock1* MUT zebrafish are largely indistinguishable from WT, even at the level of TEM analysis. Fig. S3 reveals that at P3, *Dock1* MUTs show no morphological differences other than higher g-ratios, and the thinner myelin noted in *Dock1* MUTs at P3 has resolved by P28. Fig. S4 describes the experimental design and validation of our inducible SC-specific *Dock1* MUT mouse, demonstrating that the myelin in the sciatic nerve of the icKO is indistinguishable from WT 1 mo after tamoxifen administration. Fig. S5 presents the images and quantification of WISH that illustrate an interaction between Dock1 and Rac1 in the developing zebrafish PNS.

## Data availability

Raw data underlying this study are available from the corresponding author upon request.

## Acknowledgments

*Dock1*<sup>fl/fl</sup> MUT mice were a kind gift from Jean-François Côté (Montreal Institute of Clinical Research [IRCM], Montréal, Canada). We thank Emma Brennan, Adriana Reyes, and Suhail Akram for their assistance with mouse work. We thank Austin Forbes, Adriana Reyes, and Tia Perry for caring for and maintaining our zebrafish facility. We thank Ben Emery's lab for assistance with western blotting, Peter Arthur-Farraj for his helpful advice on the nerve injury studies, and our colleagues in the Monk and Emery labs for their feedback on this manuscript. Icons in figures depicting model organisms were created in BioRender. Doan, R. (2025) https://BioRender.com/y32f649.

This work was supported by National Institutes of Health R01NS120651 to K.R. Monk and T32NS007446 to R.A. Doan.

Author contributions: R.A. Doan: conceptualization, data curation, formal analysis, funding acquisition, investigation, methodology, project administration, resources, validation, visualization, and writing—original draft, review, and editing. K.R. Monk: conceptualization, funding acquisition, methodology, project administration, supervision, validation, and writing—review and editing.

Disclosures: The authors declare no competing interests exist.

Submitted: 7 November 2023

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

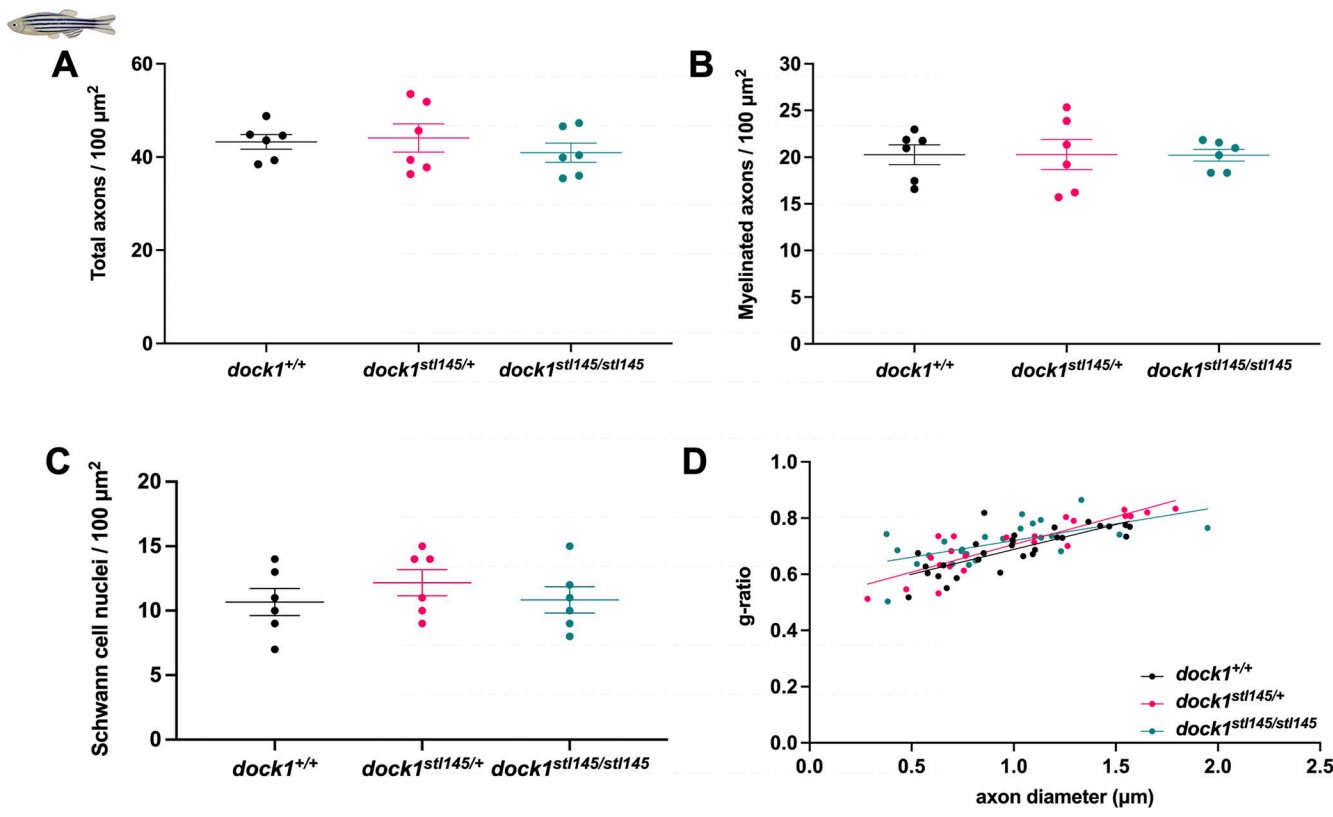

Figure S1. **dock1 MUT zebrafish do not exhibit myelin defects at 4 mo. (A–D)** Quantifications of the total number of axons, the number of myelinated axons, the number of SC nuclei, and g-ratio were obtained from analyzing TEM micrographs (see Fig. 1). None of these analyses revealed significant differences between WT, HET, or homozygous MUT zebrafish. $n$ = 6 fish per genotype. **(A–C)** One-way ANOVA with Brown–Forsythe test.

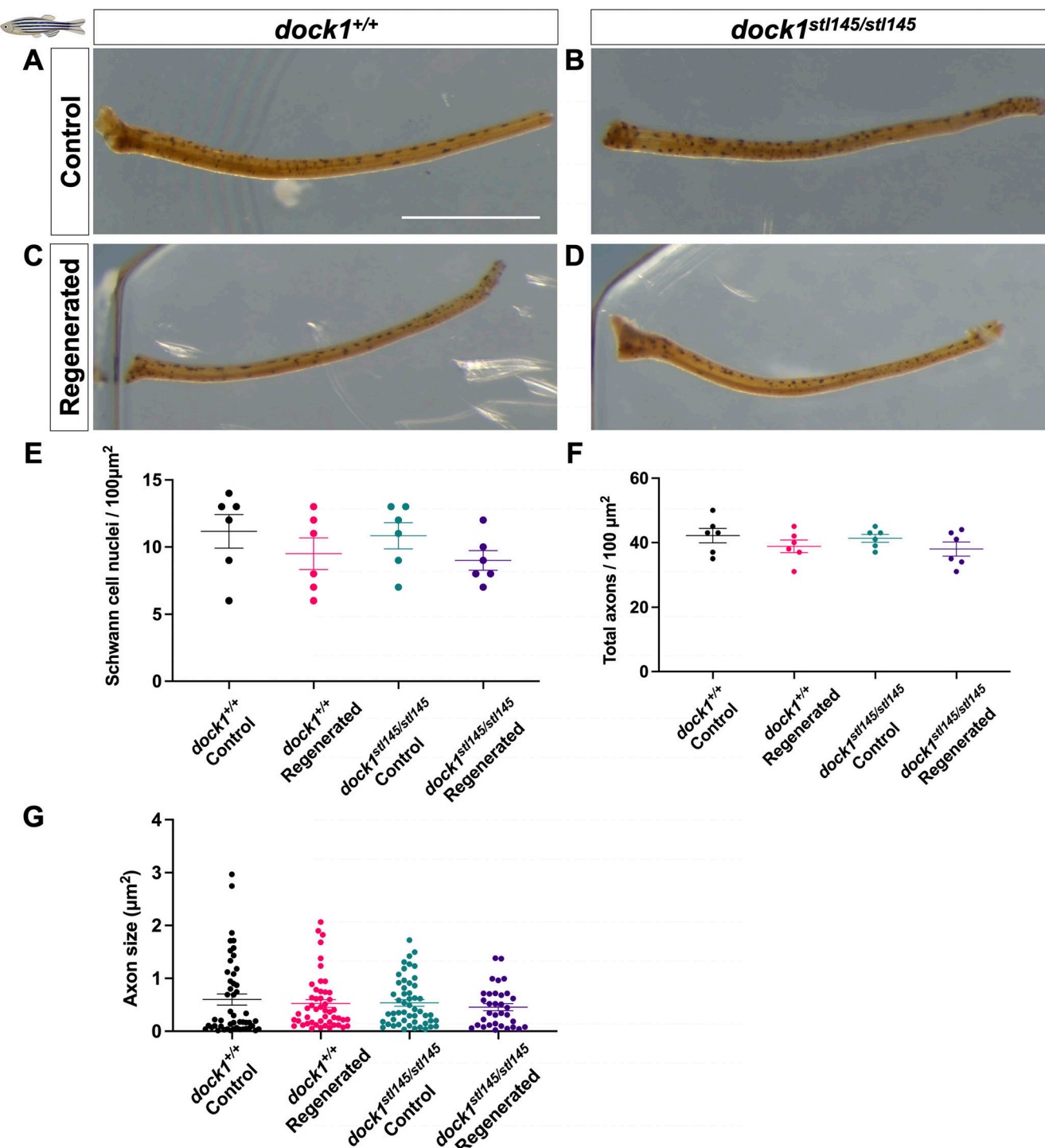

Figure S2.  **Adult *dock1* MUT zebrafish-regenerated barbels are grossly indistinguishable from WT. (A–D)** Maxillary barbels from uninjured 4-mo-old WT and *dock1* MUT zebrafish (A and B) taken from the same fish and at the same time as 28-day regenerated control and *dock1* MUTs (C and D) were harvested. **(E–G)** Quantifications from TEM micrographs (see Fig. 2) found no significant differences between genetic and experimental groups in the number of SC nuclei, total axon count, or axon size. *n* = 6 fish per genotype. **(A–D)** Scale bar = 1 mm. **(E–G)** One-way ANOVA with Brown–Forsythe test.

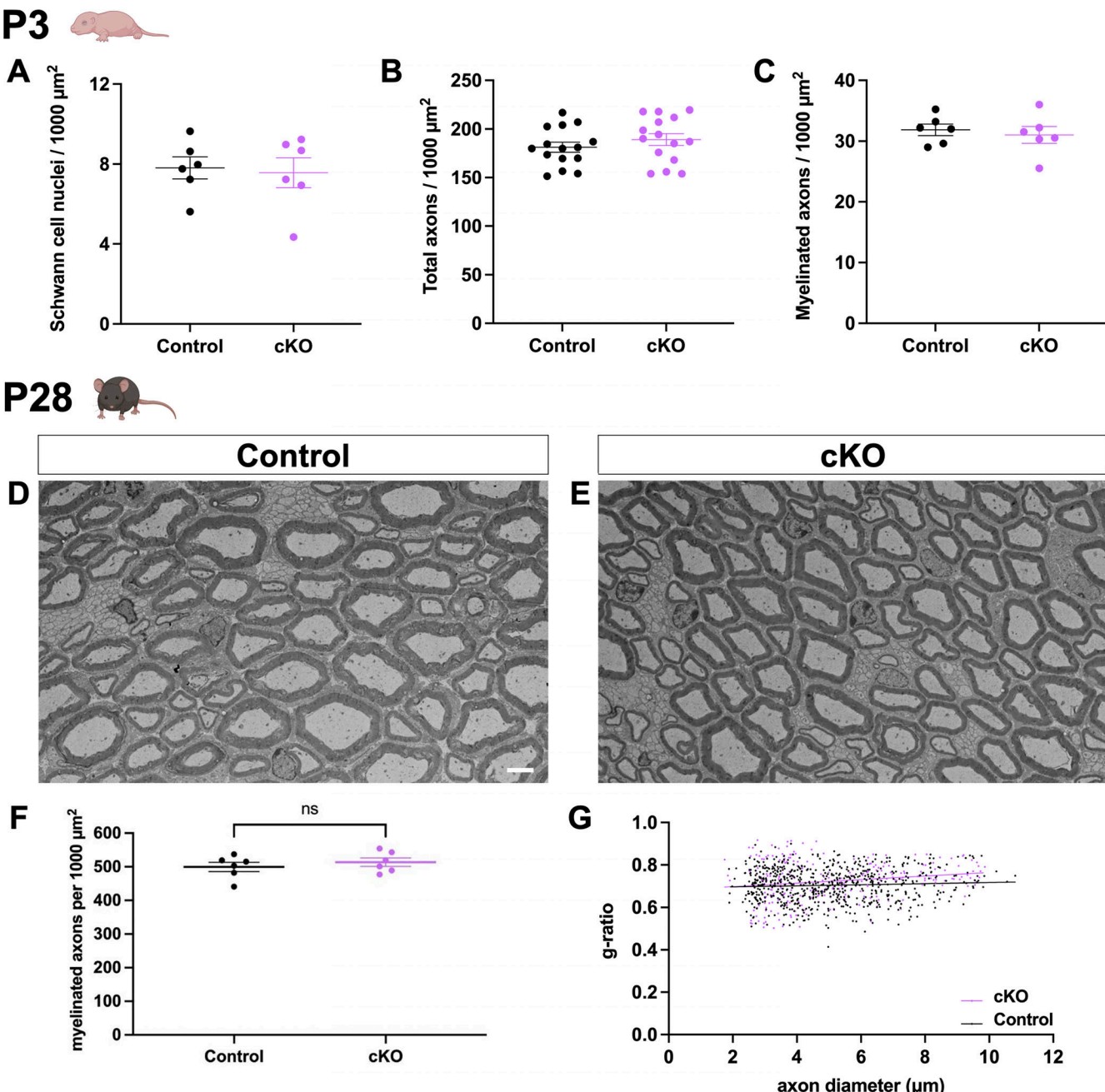

Figure S3. **The myelin phenotype observed in *Dock1* MUTs at P3 resolves by P28. (A–C)** Quantifications obtained from analyzing P3 TEM micrographs showing the number of SC nuclei, the total number of axons, and the number of myelinated axons (see Fig. 3). None of these analyses revealed significant differences between control and cKO mice at P3. **(D and E)** TEM micrographs of control and *Dock1* cKO sciatic nerves at P28. **(F and G)** Quantifications of myelinated axon count and g-ratios obtained from P28 TEM micrographs reveal no significant differences. *n* = 6 animals per genotype. **(D and E)** Scale bar = 4 μm. **(A–C and F)** Unpaired *t* test with Welch's correction. ns, not significant.

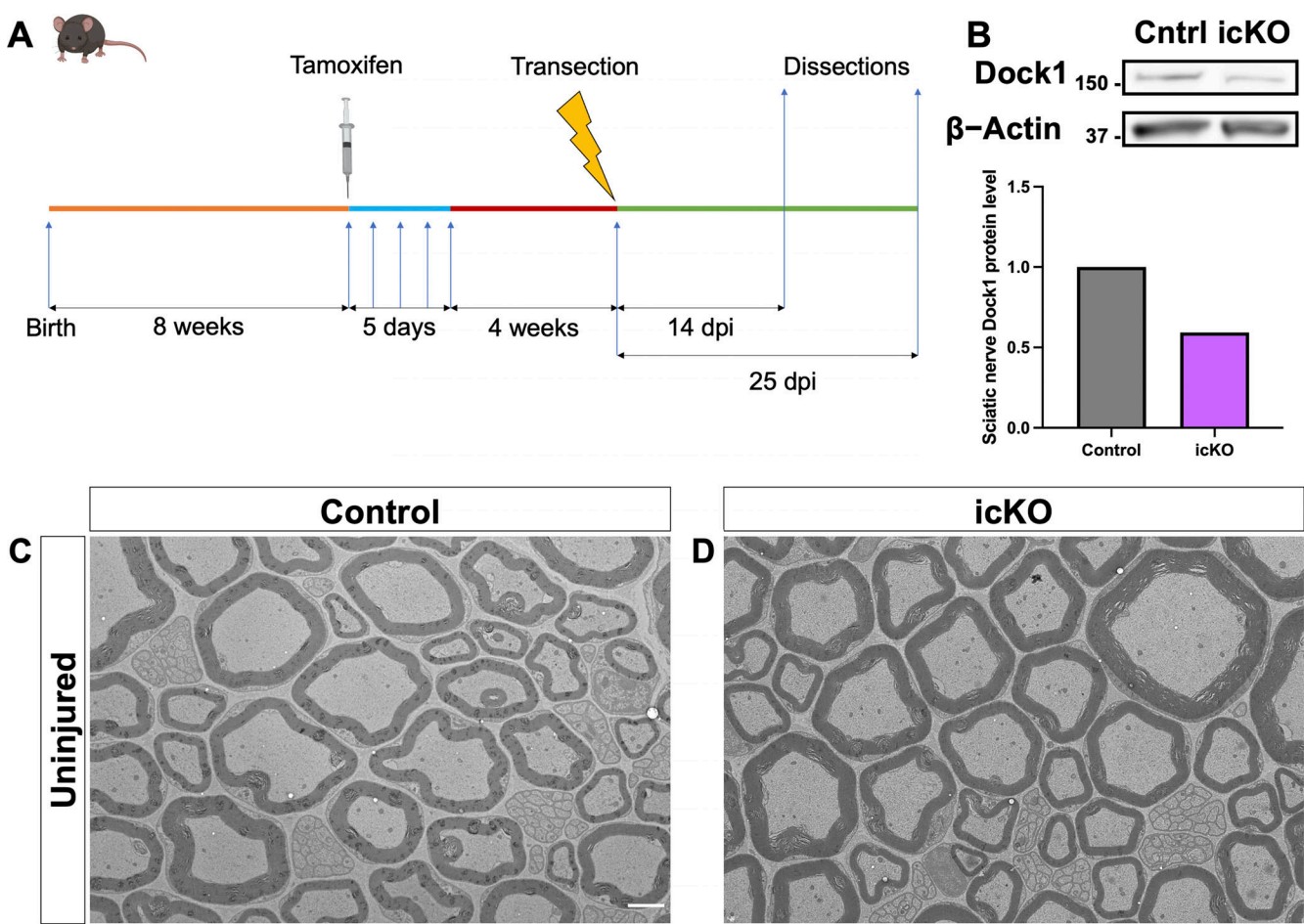

Figure S4. **An inducible SC-specific Dock1 MUT mouse to study SC repair. (A)** Schematic representation showing the experimental timeline for sciatic nerve injury studies using the icKO mice. **(B)** Western blot showing sciatic nerve Dock1 and β-actin protein levels from control and *Dock1* icKO animals and quantification of normalized protein levels. **(C and D)** TEM micrographs of sciatic nerves from control-injected and tamoxifen-injected *Plp^{Cre+};Dock1^{fl/fl}* mice before injury. **(C and D)** Scale bar = 2 μm. Source data are available for this figure: SourceData FS4.

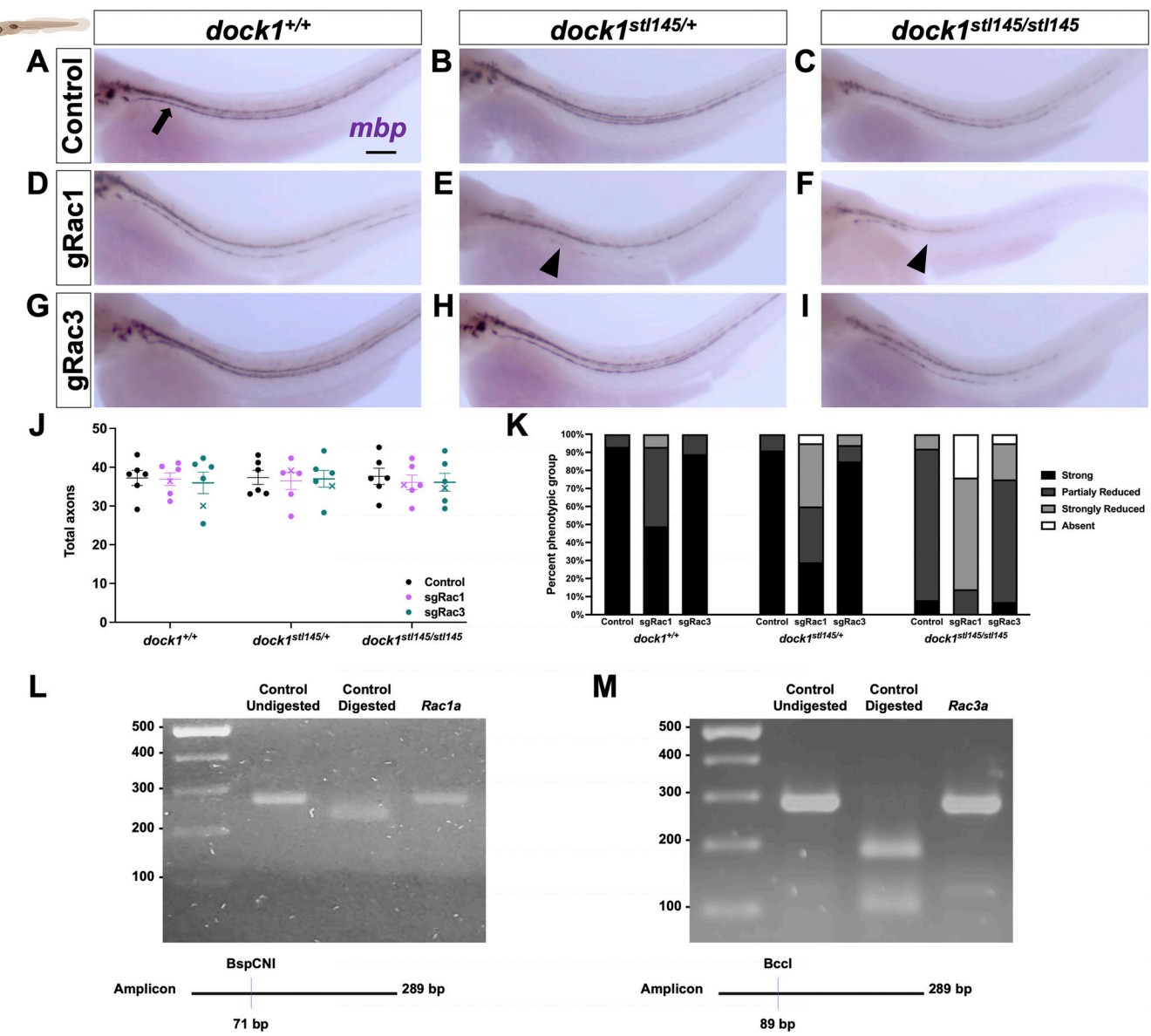

Figure S5. **WISH reveals an interaction between Dock1 and Rac1 in the developing zebrafish PNS. (A–I)** Lateral views of larvae showing *mbp* expression by WISH in control, sgRac1+Cas9, and sgRac3+Cas9 in WT *dock1*$^{+/+}$, HET *dock1*$^{stl145/+}$, and homozygous *dock1*$^{stl145/stl145}$ MUT zebrafish. **(A)** The arrow points to strong *mbp* expression in the PLLn. **(E and F)** Arrowheads highlight decreased mbp expression in the PLLn. **(J)** Quantification of TEM of total axons in the PLLn. n = 6 fish per genotype and experimental condition. **(K)** The quantification of WISH was assessed by examining *mbp* expression along the entire PLLn in 4 dpf control, sgRac1+Cas9, and sgRac3+Cas9 in WT *dock1*$^{+/+}$, HET *dock1*$^{stl145/+}$, and homozygous *dock1*$^{stl145/stl145}$ MUT zebrafish, compared between phenotypic scores and genotypes. **(L and M)** Gels and schematics of the PCR and restriction digest validation of sgRac1 and sgRac3, respectively. Here, and in all figures, the X symbol in the graph denotes a data point corresponding to the representative image shown. **(A–I)** Scale bar = 100 µm. Source data are available for this figure: SourceData FS5.

