## [Peer Review File · The Journal of Cell Biology]

Dock1 functions in Schwann cells to regulate development, maintenance, and repair

Ryan Doan and Kelly Monk

Corresponding Author(s): Kelly Monk, Oregon Health & Science University

Review Timeline:

Submission Date:	2023-11-07
Editorial Decision:	2023-12-20
Revision Received:	2024-12-05
Editorial Decision:	2025-01-21
Revision Received:	2025-02-04

Monitoring Editor: Elior Peles

Scientific Editor: Tim Spencer

Transaction Report:

DOI: <https://doi.org/10.1083/jcb.202311041>

December 19, 2023

Re: JCB manuscript #202311041

Dr. Kelly R Monk
Oregon Health & Science University
Vollum Institute
3181 SW Sam Jackson Park Rd
Vollum 3431A
PORTLAND, OR 97239

Dear Kelly,

Thank you for submitting your manuscript entitled "Dock1 functions in Schwann cells to regulate development, maintenance, and repair". The manuscript has been evaluated by expert reviewers, whose reports are appended below. Unfortunately, after an assessment of the reviewer feedback, our editorial decision is against publication in JCB.

You will see that, although the referees find the premise of the study potentially interesting, reviewers #1 and #2 feel that the advance represented by the study does not quite reach the level expected of a JCB Report. These reviewers also feel (in their reports and in private comments to the editors) that the depth of mechanistic insight provided in the study would not make the paper suitable as a full Article either. I am afraid that we agree with these assessments.

Thus, although your manuscript is intriguing, we feel that the points raised by the reviewers are more substantial than can be addressed in a typical revision period. If you wish to expedite publication of the current data, it may be best to pursue publication at another journal.

Given interest in the topic, though, if you were to extend the study to provide new insight into the mechanisms by which Dock1 promotes myelin maintenance and regeneration (as well as addressing the other issues raised by the reviewers), we would be open to an appeal of our decision and resubmission to JCB as a full Article. If you would like to resubmit this work to JCB, please contact the journal office to discuss an appeal of this decision or you may submit an appeal directly through our manuscript submission system. However, we realize that this would require substantial time and resources and represent a considerable extension of the study and you may not wish to pursue that avenue at this time. Therefore, if you wish to get this paper published as soon as possible, we would be happy to transfer the reviewer reports to any other journal of your choosing.

To that end, we encourage you to transfer your study to our sister journal, Life Science Alliance (LSA). LSA is an academic editor-led, not-for-profit open-access journal launched as a collaboration between RUP, EMBO Press and Cold Spring Harbor Press.

We shared your manuscript and the accompanying reviews with LSA Executive Editor, Eric Sawey, who is interested in these findings, and would like to invite further consideration of this manuscript at LSA pending the following revisions:

- Address Reviewer 1's questions regarding how the transection was performed.
- Address Reviewer 2's points #2 & 4, as well as the Minor points. Point #3 can be addressed via added Discussion.
- Address Reviewer 3's minor point.

You do not need to revise the manuscript before transferring it to LSA. Once you transfer, Dr. Sawey will email you an invitation to revise and resubmit, listing the same revision requests as mentioned above. Please feel free to reach out at e.sawey@life-science-alliance.org if you have any questions about the LSA journal, the transfer process or the revisions requested.

If you wish to pursue this option, please let us know at cellbio@rockefeller.edu.

Regardless of how you choose to proceed, we hope that the comments below will prove constructive as your work progresses.

Thank you for thinking of JCB as an appropriate place to publish your work.

Sincerely,

Elior Peles, PhD
Monitoring Editor
Journal of Cell Biology

Tim Spencer, PhD

Reviewer #1 (Comments to the Authors (Required)):

The manuscript by Doan and Monk describes a collection of experiments examining the role of the guanine nucleotide exchange factor (GEF) in myelin maintenance and in remyelination following traumatic injury to the nerve. In previous studies, this lab has shown the Dock1 is required for correct developmental timing of early myelination but that any delayed myelination is completely resolved quickly at later postnatal stages. Here they use zebrafish and mouse models to demonstrate that in the absence of Dock1, myelin maintenance is compromised with myelin abnormalities appearing at late adult life. They further demonstrate that Dock1 mutation also affects remyelination following nerve transection. Using the zebrafish Dock1 mutant and wildtype fish, they demonstrate a potential interaction between Dock1 and Rac1 in regulating developmental myelination by using a pharmacological compound that inhibits rac1 activity.

The experiments are well-designed and controlled and group sizes allow reliable statistical treatment of the data. The data support the main conclusions of the study and the reported observations are of interest.

The authors might want to address the following points:

In the barbel regeneration model, the barbel is completely removed and allowed to regenerate for several weeks before the barbel is dissected and analysed. In this case, axonal regeneration is accompanied by migrating Schwann cells that most likely originate from injury-induced repair Schwann cells at the point of transection. This requires these Schwann cells to proliferate, generate a basal lamina, sort and migrate with the growing axon before they will remyelinate the regenerated axons. Thus, this model recapitulates more aspects of Schwann cell biology than the more traditional transection injury model (in which axons regrow into bands of Bungers formed by repair Schwann cells). Therefore, a conclusion that Dock1 plays a role in remyelination is unwarranted based on these experiments alone.

The classical transection model in mouse is more straightforward in this respect. However, it is not entirely clear to me from the description in the material and methods, how the transection was performed. It is stated that the nerve is transected and the wound closed. How was regeneration of the proximal stump into the distal stump optimised? Were the nerve segments placed in proximity by a collar? If not, the extent of regrowth of axons into the distal nerve stump will be very variable complicating the interpretation of the images of regenerated fibers in the distal nerve stump. A clearer description is needed to fully appreciate the details of the experiment presented in figure 5.

The conclusion that Dock1 interacts with Rac1 is solely based on the pharmacological inhibition of Rac1 with compound EHT1864. How specific is this compound for Rac1 or does it also inhibit other GTPases such as cdc42?

Reviewer #2 (Comments to the Authors (Required)):

This study examines the role of Dock1, a GEF for Rac1, in PNS myelination. The authors had previously shown a mutation of Dock1 delays PNS myelination in zebrafish. They now show Dock1 is required to maintain PNS myelin and to promote effective regeneration, that it acts in a Schwann cell autonomous fashion, and provide evidence that it functions via Rac1 signaling. The findings are generally convincing and well-presented. A strength of the study is the combined use of the pan-zebrafish knockout together with a newly generated Schwann cell specific, mouse cKO. However, evidence that the effects of Dock1 are mediated specifically by Rac1 could be stronger. In addition, the mechanistic advance over the prior study is modest and would benefit from additional analyses. Specific issues include:

1. The Dock1 cKOs do not fully phenocopy the published Rac1 cKOs as they have a milder dysmyelination phenotype that resolves much more quickly. Evidence that Dock1 mutants specifically alters Rac1 signaling is indirect and relies on pharmacological inhibition of Rac exacerbating this mild phenotype of the Dock1 het. Direct evidence that the Dock1 mutation impacts Rac1 activation by examining the Rac1-GTP bound fraction (vs. total Rac1) in the cKO nerves would strengthen this important point. As published data suggests the effects of Dock1 should be Rac-specific, cdc42 activation would be a useful comparison/control. A related issue is raised by a recent study (<https://doi.org/10.1016/j.neulet.2021.135868>) demonstrating that a Rac3 KO temporarily ameliorates the myelination defects of Rac1 KO initially but at a later time point (at 2 months) this amelioration is lost (i.e. myelin maintenance is impaired). This phenotype raises the possibility that Dock1 KO impacts Rac3 activation and results in a compound Rac phenotype. Thus analysis of Rac3 activation (GTP-bound fraction), if feasible, would be of interest as well. While the inhibitor is reported to have a higher affinity for Rac1 than Rac3 it is possible Rac3 is nevertheless impacted. Analysis of GTP bound Rac1 and Rac3 in the presence of the inhibitor would also be of interest.

2. The impairment of regeneration is convincing but the characterization of the defect is limited particularly in the mouse nerve transection. The distal site appears less cellular and there are more foamy macrophages, but this was not quantified. Is this a deficit entirely of SCs or are the numbers of macrophages impacted? Are there more foamy macrophages because there are fewer SC available for autophagocytosis? Is SCs expression of c-Jun impacted? Some analysis with cellular markers to assess the hypocellularity (and proliferation assays if confirmed) may point to whether defects of regeneration result from limited

numbers of Schwann cells, macrophages, and/or altered Schwann cell differentiation.

3. The EMs convincingly show a variety of abnormalities in the 12 month-old nerves but how these might result from altered Dock1/Rac signaling is unclear and not discussed explicitly. Defects are also seen in Remak fibers, not just myelinating Schwann cells, broadening the nature of the defects. Is there demyelination with remyelination and are these late defects recapitulating early defects of ensheathment and myelination?

4. The delay in myelination and its resolution is shown in mouse sciatic nerve but not in the ZMB nerves of the zebrafish. They show myelination is normal at 4 months in the ZMB system but they don't show it is abnormal at an earlier time of myelination e.g., about 1 month when this nerve starts to myelinate. (They do show it is delayed in PLL at an early time point but not that it resolves at late time points in the PLL)

Minor points the authors should consider:

- The authors might consider reorganizing Figure 6 (panels Fig. 6G and H) to show side by side the effects of drug on each genotype (wt, +/-, and -/-) as 3 pairs of bar graphs located in a row in between the corresponding panels above (A-F) and below (I-N). This would facilitate comparisons of the effects of the drug on each phenotype

- Is the change in myelinated axon profiles seen in the TEMs in Fig. 1 from circular to flattened profiles at 4 vs 12 month, respectively meaningful or due to technical variables such as quality of fixation

- Do the regeneration defects improve over time - they only report the 1 month time point

Reviewer #3 (Comments to the Authors (Required)):

The manuscript by Doan and Monk entitled "Dock1 functions in Schwann cells to regulate development, maintenance, and repair" provides a comprehensive characterization of the evolutionarily conserved role of Dock1 in Schwann cell myelination in development, myelin maintenance in aged animals as well as in remyelination after injury.

In this report, Doan and Monk expand our knowledge on the role of Dock1 in Schwann cell biology, an important GEF that they originally demonstrated to be required for radial sorting and developmental PNS myelination in Cunningham et al., 2018.

Here, the authors developed an unconventional yet powerful and elegant model of adult peripheral remyelination: the zebrafish maxillary barbel (ZMB) amputation assay. They also investigate the role of Dock1 in mammalian Schwann cell developmental myelination, maintenance, and remyelination after sciatic nerve transection using conditional knockout mouse models.

The manuscript is very well written, and the experimental layout is logical and convincing. The description of the statistical analysis and sample size is irreproachable. Along these lines, I wanted to stress that I appreciate the inclusion of dock1^{stl145/+} heterozygous data in the main findings.

I support the publication of this manuscript as a JCB Report as it is, and I only have the following minor point for the authors:

- In the zebrafish nerve injury regeneration assay (Figure 2D-E), dock1^{stl145/stl145} regenerated nerves present a dramatic decrease in the number of myelinated axons in the ZMB per 100 μ m², as well as a decrease in myelin thickness. While the authors discuss ZMB regeneration in their length, the number of Schwann cells and axons (Fig S2), they do not describe any change in axon caliber. Although the TEM micrograph in figure 2D shows axons of various diameters, axonal diameter in the regenerated ZMB is not discussed. Adding a graph showing the distribution of axon diameters in the intact vs regenerated dock1^{stl145/stl145} ZMB to Figure 2 or Figure S2, would clarify the results and would alleviate the concern that the myelin phenotype is the direct consequence of a decrease in axonal diameter in the regenerated ZMB.

JCB manuscript #202311041

Response to referees:

General comment: We thank the Reviewers for their time in assessing our manuscript and for the very helpful suggestions and points for clarification. Below, we provide responses to all points raised in the first round of review. We hope the Reviewers will agree that we have adequately responded to all major concerns.

Reviewer #1

1. In the barbel regeneration model, the barbel is completely removed and allowed to regenerate for several weeks before the barbel is dissected and analyzed. In this case, axonal regeneration is accompanied by migrating Schwann cells that most likely originate from injury-induced repair Schwann cells at the point of transection. This requires these Schwann cells to proliferate, generate a basal lamina, sort and migrate with the growing axon before they will remyelinate the regenerated axons. Thus, this model recapitulates more aspects of Schwann cell biology than the more traditional transection injury model (in which axons regrow into bands of Bungers formed by repair Schwann cells). Therefore, a conclusion that Dock1 plays a role in remyelination is unwarranted based on these experiments alone.

Response:

We appreciate this comment and agree the ZBM transection model is more complicated than a traditional nerve cut or crush, and in some ways models aspects of development or even limb regeneration more than a more straightforward nerve injury. We devoted significant time attempting to establish a crush model in the ZBMs as well as tail fins; unfortunately, every attempt snapped the fragile tissues. We agree with the reviewer that a conclusion regarding Dock1's role in remyelination cannot be made from the ZBM experiments alone. Given that we were unable to establish a ZBM crush model, we have modified the language for this section's subheading (page 6, line 121) and conclusion sentence (page 7, line 142), as well as the corresponding figure subtitle (Figure 2).

2. The classical transection model in mouse is more straightforward in this respect. However, it is not entirely clear to me from the description in the material and methods, how the transection was performed. It is stated that the nerve is transected and the wound closed. How was the regeneration of the proximal stump into the distal stump opted? Were the nerve segments placed in proximity by a collar? If not, the extent of regrowth of axons into the distal nerve stump will be very variable complicating the interpretation of the images of regenerated fibers in the distal nerve stump. A clearer description is needed to fully appreciate the experiment details presented in figure 5.

Response: We have rewritten the methods section related to the sciatic nerve transection, adding detail to provide a more precise account of how the surgery was performed (pages 25-26, lines 561-577).

3. The conclusion that Dock1 interacts with Rac1 is solely based on the pharmacological inhibition of Rac1 with compound EHT1864. How specific is this compound for Rac1 or does it also inhibit other GTPases such as cdc42?

Response: EHT1864 is highly selective for Rac1 (K_D 40 nM) although it can also inhibit Rac3 to a lesser degree (K_D 230 nM); importantly it does not affect RhoA or Cdc42 (Onesto et al., 2008; Shutes et al., 2007). This information and citations have been added to the manuscript (page 11, lines 256-259).

To strengthen the mechanistic link between Dock1 and Rac1, we also performed new experiments. First, we performed a genetic interaction study in zebrafish which Rac1 is targeted in *dock1* mutants using CRISPR/Cas9-mediated genome editing. We show that targeting *rac1* enhanced homozygous mutant phenotypes while also introducing phenotypes into *dock1* heterozygous animals, which are normally indistinguishable from WT. Importantly, targeting *rac3* had no such effect. These experiments and data have been added to the revised manuscript (page 13-15, lines 297-331; page 17, lines 429-432; page 32-33, lines 732-748; Figure 7A-J; Supplemental Figure 5). We also assessed levels of active, GTP-bound Rac1 in conditional mutant mouse sciatic nerve and by pulldown assays and found that levels of active Rac1 in *Dock1* mutant nerves are significantly decreased relative to wild-type animals (page 15, lines 333-338; page 17, lines 429-432; page 33-34, lines 750-761; Figure 7K).

Reviewer #2

1. The Dock1 cKOs do not fully phenocopy the published Rac1 cKOs as they have a milder dysmyelination phenotype that resolves much more quickly. Evidence that Dock1 mutants specifically alters Rac1 signaling is indirect and relies on pharmacological inhibition of Rac exacerbating this mild phenotype of the Dock1 het. Direct evidence that the Dock1 mutation impacts Rac1 activation by examining the Rac1-GTP bound fraction (vs. total Rac1) in the cKO nerves would strengthen this important point.

We thank the reviewer for this suggestion and performed the experiment. Please see response to Reviewer 1, point 3.

As published data suggests the effects of Dock1 should be Rac-specific, *cdc42* activation would be a useful comparison/control. A related issue is raised by a recent study (<https://doi.org/10.1016/j.neulet.2021.135868>) demonstrating that a Rac3 KO temporarily ameliorates the myelination defects of Rac1 KO initially but at a later time point (at 2 months) this amelioration is lost (i.e. myelin maintenance is impaired). This phenotype raises the possibility that Dock1 KO impacts Rac3 activation and results in a compound Rac phenotype. Thus analysis of Rac3 activation (GTP-bound fraction), if feasible, would be of interest as well. While the inhibitor is reported to have a higher affinity for Rac1 than Rac3 it is possible Rac3 is nevertheless impacted. Analysis of GTP bound Rac1 and Rac3 in the presence of the inhibitor would also be of interest.

We performed genetic interaction studies in zebrafish to address the potential contributions of Rac3; please see response to Reviewer 1, point 3. We also modified the text to clarify the specificity of EHT1864 (Reviewer 1, point 3) as well as to cite and discuss the Rac3/Rac1 study that the Reviewer pointed out (page 14-15, lines 321-323).

2. The impairment of regeneration is convincing but the characterization of the defect is limited particularly in the mouse nerve transection. The distal site appears less cellular and there are more foamy macrophages, but this was not quantified. Is this a deficit

entirely of SCs or are the numbers of macrophages impacted? Are there more foamy macrophages because there are fewer SC available for autophagocytosis? Is SCs expression of c-Jun impacted? Some analysis with cellular markers to assess the hypocellularity (and proliferation assays if confirmed) may point to whether defects of regeneration result from limited numbers of Schwann cells, macrophages, and/or altered Schwann cell differentiation.

We performed new analyses to more fully characterize defects underlying impaired regeneration in the absence of *Dock1*. We quantified electron micrographs and found that at 14 days-post injury (dpi), mutant nerves had significantly more macrophages containing intact myelin debris, more foamy macrophages, and more presumptive motile macrophages altogether suggesting delayed debris clearance and delayed and ongoing macrophage recruitment. At 25 dpi, mutant nerves had significantly fewer myelinated axons and the myelin that was present was thinner than controls (although the number of large axons was not different). Additionally, we also found persistent evidence of delayed debris clearance at 25 dpi as macrophages in mutant nerves contained more visible lipid droplets than control. These data have been added to the revised manuscript (page 11, lines 228-249; page 18, lines 405-414; Figure 5).

3. The EMs convincingly show a variety of abnormalities in the 12 month-old nerves but how these might result from altered Dock1/Rac signaling is unclear and not discussed explicitly. Defects are also seen in Remak fibers, not just myelinating Schwann cells, broadening the nature of the defects. Is there demyelination with remyelination and are these late defects recapitulating early defects of ensheathment and myelination?

Response: We expanded the discussion to include these important points (page 17-18, lines 385-393).

4. The delay in myelination and its resolution is shown in mouse sciatic nerve but not in the ZMB nerves of the zebrafish. They show myelination is normal at 4 months in the ZMB system but they don't show it is abnormal at an earlier time of myelination e.g., about 1 month when this nerve starts to myelinate. (They do show it is delayed in PLL at an early time point but not that it resolves at late time points in the PLL)

Response: We attempted to examine ZBMs at an earlier time points as suggested; however, even as late as 8 weeks of age, these structures were not long enough for us to remove and process. We did not examine later stages of PLL nerve phenotypes than originally presented (Cunningham et al., 2018) since for this nerve, the presented stages with phenotypes were relevant to the present study and our genetic and pharmacological interaction studies. It would be difficult to extrapolate meaning across two different nerves (ZMB and PLL nerves), which develop at distinct life stages and serve very different functions. Moreover, it would be difficult to conclude much about Schwann cell-specific functions from the global mutant zebrafish. In this way, we think that the time-course analyses performed in conditional mutant mouse sciatic nerve provide more interpretable data.

Minor points the authors should consider:

5. The authors might consider reorganizing Figure 6 (panels Fig. 6G and H) to show side by side the effects of drug on each genotype (wt, +/-, and -/-) as 3 pairs of bar graphs located in a row in between the corresponding panels above (A-F) and below (I-N). This would facilitate comparisons of the effects of the drug on each phenotype

Response: We appreciate this suggestion and have redesigned the prior data to be represented by a single graph (Figure 6G) that better illustrates the effect of the drug on each genotype.

6. Is the change in myelinated axon profiles seen in the TEMs in Fig. 1 from circular to flattened profiles at 4 vs 12 month, respectively meaningful or due to technical variables such as quality of fixation

Response: This lack of circularity is not uncommon in zebrafish myelinated axons, and we believe they are the result of technical, not biological issues. To avoid confusion, we have replaced the previous images with other images (already used as part of the quantification) in order to show more similar axon circularity across figure panels.

7. Do the regeneration defects improve over time - they only report the 1 month time point

Response: While we have not done extensive analysis of this, in a pilot experiment where we re-transected a regenerated barbel to look at a double injury model, we saw that regenerated barbels in mutant fish were indistinguishable from wild-type fish 4 months after the initial transection, suggesting regeneration defects do improve with time.

Reviewer #3

1. In the zebrafish nerve injury regeneration assay (Figure 2D-E), dock1stl145/stl145 regenerated nerves present a dramatic decrease in the number of myelinated axons in the ZMB per 100um², as well as a decrease in myelin thickness. While the authors discuss ZMB regeneration in their length, the number of Schwann cells and axons (Fig S2), they do not describe any change in axon caliber. Although the TEM micrograph in figure 2D shows axons of various diameters, axonal diameter in the regenerated ZMB is not discussed. Adding a graph showing the distribution of axon diameters in the intact vs regenerated dock1stl145/stl145 ZMB to Figure 2 or Figure S2, would clarify the results and would alleviate the concern that the myelin phenotype is the direct consequence of a decrease in axonal diameter in the regenerated ZMB.

Response: As suggested by the reviewer, we quantified the axon size in the control and regenerated barbels for both genotypes and found no significant differences. These data are now included in the revised manuscript (page 7, lines 139-142, Supplemental Figure 2).

References

- Cunningham, R.L., A.L. Herbert, B.L. Harty, S.D. Ackerman, and K.R. Monk. 2018. Mutations in dock1 disrupt early Schwann cell development. *Neural Dev.* 13:17.
- Onesto, C., A. Shutes, V. Picard, F. Schweighoffer, and C.J. Der. 2008. Characterization of EHT 1864, a novel small molecule inhibitor of Rac family small GTPases. *Methods Enzymol.* 439:111-129.

Shutes, A., C. Onesto, V. Picard, B. Leblond, F. Schweighoffer, and C.J. Der. 2007. Specificity and mechanism of action of EHT 1864, a novel small molecule inhibitor of Rac family small GTPases. *J Biol Chem.* 282:35666-35678.

January 21, 2025

RE: JCB Manuscript #202311041R-A

Kelly Monk
Oregon Health & Science University

Dear Kelly:

Thank you for submitting your revised manuscript entitled "Dock1 functions in Schwann cells to regulate development, maintenance, and repair". Your paper has now been seen again by two of the original reviewers, both of whom recommend acceptance. Therefore, we would be happy to publish your paper in JCB pending final revisions necessary to meet our formatting guidelines (see details below).

****Reviewer #2 has raised three final concerns that we would like you to address in the final revision. These seem straightforward and should only require minor modifications of the figures and text. Please be sure to include a brief point-by-point rebuttal which illustrates how these issues were addressed.****

A. MANUSCRIPT ORGANIZATION AND FORMATTING:

1) Text limits: Character count for Articles is normally < 40,000, not including spaces. Count includes the abstract, introduction, results, discussion, and acknowledgments. Count does not include title page, materials and methods, figure legends, references, tables, or supplemental legends.

Needless to say, your paper exceeds this limit. However, given that significant amounts of text were added during revision in response to reviewer queries, we will be able to give you the extra space this time, but please try not to add substantially to the manuscript length in the final revision, if you can help it.

2) Figure formatting: Scale bars must be present on all microscopy images, including inset magnifications. Molecular weight or nucleic acid size markers must be included on all gel electrophoresis. Even though the blots in figures 3A, 7K, and S4B are highly cropped images of single bands, we still require at least one indicator of molecular weight so please add molecular weight markers to these three blots.

3) Statistical analysis: Error bars on graphic representations of numerical data must be clearly described in the figure legend. The number of independent data points (n) represented in a graph must be indicated in the legend. Statistical methods should be explained in full in the materials and methods. For figures presenting pooled data the statistical measure should be defined in the figure legends. Please also be sure to indicate the statistical tests used in each of your experiments (both in the figure legend itself and in a separate methods section) as well as the parameters of the test (for example, if you ran a t-test, please indicate if it was one- or two-sided, etc.).

****Also, since you used parametric tests in your study (e.g. t-tests, ANOVA, etc.), you should have first determined whether the data was normally distributed before selecting that test. In the stats section of the methods, please indicate how you tested for normality. If you did not test for normality, you must state something to the effect that "Data distribution was assumed to be normal but this was not formally tested."****

4) Materials and methods: Should be comprehensive and not simply reference a previous publication for details on how an experiment was performed. Please provide full descriptions (at least in brief) in the text for readers who may not have access to referenced manuscripts. The text should not refer to methods "...as previously described."

5) Please be sure to provide the sequences for all of your primers/oligos and RNAi constructs in the materials and methods. You must also indicate in the methods the source, species, and catalog numbers (where appropriate) for all of your antibodies.

6) Microscope image acquisition: The following information must be provided about the acquisition and processing of images:

- a. Make and model of microscope
- b. Type, magnification, and numerical aperture of the objective lenses
- c. Temperature
- d. imaging medium
- e. Fluorochromes
- f. Camera make and model

g. Acquisition software

h. Any software used for image processing subsequent to data acquisition. Please include details and types of operations involved (e.g., type of deconvolution, 3D reconstitutions, surface or volume rendering, gamma adjustments, etc.).

7) References: There is no limit to the number of references cited in a manuscript. References should be cited parenthetically in the text by author and year of publication. Abbreviate the names of journals according to PubMed.

8) Supplemental materials: There are strict limits on the allowable amount of supplemental data. Articles/Tools may have up to 5 supplemental figures. At the moment, you are below this limit but please bear it in mind when revising.

Please also note that tables, like figures, should be provided as individual, editable files. A summary of all supplemental material (that is, in addition to the supplementary figure legends) should appear at the end of the Materials and methods section. Please see any recent JCB paper for an example of this.

9) eTOC summary: A ~40-50 word summary that describes the context and significance of the findings for a general readership should be included on the title page. We realize that you already provided one in our online system, but we recommend that the statement be written in the present tense and refer to the work in the third person. It should contain "First author name(s) et al..." (or, in your case, "Doan and Monk...") to match our preferred style.

10) Conflict of interest statement: JCB requires inclusion of a statement in the acknowledgements regarding competing financial interests. If no competing financial interests exist, please include the following statement: "The authors declare no competing financial interests." If competing interests are declared, please follow your statement of these competing interests with the following statement: "The authors declare no further competing financial interests."

11) A separate author contribution section is required following the Acknowledgments in all research manuscripts. All authors should be mentioned and designated by their first and middle initials and full surnames. We encourage use of the CRediT nomenclature (<https://casrai.org/credit/>).

12) ORCID IDs: ORCID IDs are unique identifiers allowing researchers to create a record of their various scholarly contributions in a single place. Please note that ORCID IDs are now *required* for all authors. At resubmission of your final files, please be sure to provide your ORCID ID and those of all co-authors.

13) Journal of Cell Biology now requires a data availability statement for all research article submissions. These statements will be published in the article directly above the Acknowledgments. The statement should address all data underlying the research presented in the manuscript. Please visit the JCB instructions for authors for guidelines and examples of statements at (<https://rupress.org/jcb/pages/editorial-policies#data-availability-statement>).

B. FINAL FILES:

****It is JCB policy that if requested, original data images must be made available to the editors. Failure to provide original images upon request will result in unavoidable delays in publication. Please ensure that you have access to all original data images prior to final submission.****

****The license to publish form must be signed before your manuscript can be sent to production. A link to the electronic license to publish form will be sent to the corresponding author only. Please take a moment to check your funder requirements before choosing the appropriate license.****

Thank you for your attention to these final processing requirements. Please revise and format the manuscript and upload materials within 7-14 days. If you need an extension for whatever reason, please let us know and we can work with you to determine a suitable revision period.

Thank you for this interesting contribution, we look forward to publishing your paper in Journal of Cell Biology.

Sincerely,

Elior Peles
Monitoring Editor
Journal of Cell Biology

Tim Spencer, PhD
Executive Editor
Journal of Cell Biology

Reviewer #1 (Comments to the Authors (Required)):

The authors responded adequately to my initial concern that the Barbel regeneration model in zebrafish cannot be directly compared to the nerve transection model in mouse. In conjunction they have provided a more detailed description of the nerve transection experiments in mice. Importantly, the authors now provide more evidence to suggest that Dock1 indeed activates Rac1. I appreciate the addition of the western blot showing that active Rac1 levels are reduced in the absence of Dock1.

Reviewer #2 (Comments to the Authors (Required)):

This is a revised manuscript that addresses the role of Dock1 in myelin maintenance and repair by Schwann cells. It is technically very strong and clearly presented. The authors have been very responsive and addressing concerns raised by the reviewers, including adding new data that convincingly links loss of Dock1 to reduced Rac1 activity. This has strengthened the study and together the changes advances this report over their earlier study linking Dock1 to myelin formation. I have a few minor comments the authors should clarify

1. In the measurement of active/total Rac1 in figure 7K, they show bar graphs but no SEM; consequently, it is unclear if this measurement was done once or multiple times and is statistically significant. If done multiple times, they should show the data as scatter plots as they have done for all the other data in the MS
2. in the analysis of MBP by WISH (Fig. 6G, SFig. 5), they classify staining by the degree of MBP reduction. The authors should clarify whether this analysis corresponds to the extent of staining in the same region of an individual nerve (different HPFs?), different sites on the same nerve, or the same site on different nerves, and what the n's were for this analysis. In Fig. 6 A-F, the arrow and arrowheads, which appear to demarcate the PLL (distinguishing it from the more darkly stained spinal cord), should be described
3. The Fig. 5 legend states "Need to describe F". They still need to do so.

31 January 2025

Re: JCB Manuscript #202311041R-A

Dear Dr. Spencer:

Thank you for your continued time in coordinating the review of our revised manuscript "Dock1 functions in Schwann cells to regulate development, maintenance, and repair." We are delighted that the work will be published in JCB.

Here, we respond to Reviewer #2's three final concerns as well as a couple of organization/formatting points raised in the acceptance email from 21 January 2025.

Reviewer #2 Final Comments:

1. In the measurement of active/total Rac1 in figure 7K, they show bar graphs but no SEM; consequently, it is unclear if this measurement was done once or multiple times and is statistically significant. If done multiple times, they should show the data as scatter plots as they have done for all the other data in the MS

The experiment was performed twice with n=6 animals (12 nerves) per genotype each time. Unfortunately, technical issues with the imager precluded data capture for one technical replicate. Thus, the data shown in Figure 7 are from one representative technical replicate of 6 animals/12 nerves per genotype. Figure 7's legend and the methods have been updated to clarify this.

2. in the analysis of MBP by WISH (Fig. 6G, SFig. 5), they classify staining by the degree of MBP reduction. The authors should clarify whether this analysis corresponds to the extent of staining in the same region of an individual nerve (different HPFs?), different sites on the same nerve, or the same site on different nerves, and what the n's were for this analysis. In Fig. 6 A-F, the arrow and arrowheads, which appear to demarcate the PLL (distinguishing it from the more darkly stained spinal cord), should be described

We updated the legends for Figure 6 and Supplemental Figure 5 as well as the methods to clarify that we analyzed *mbp* levels of the entire the lateral line. We have also included details in the corresponding figure legends to better explain what the arrows and arrowheads highlight.

3. The Fig. 5 legend states "Need to describe F". They still need to do so.

Sincere apologies for missing this in the last submission. This legend has been updated.

A. MANUSCRIPT ORGANIZATION AND FORMATTING:

1) Text limits: Character count for Articles is normally < 40,000, not including spaces. Count includes the abstract, introduction, results, discussion, and acknowledgments. Count does not include title page, materials and methods, figure legends, references, tables, or supplemental legends.

Needless to say, your paper exceeds this limit. However, given that significant amounts of text were added during revision in response to reviewer queries, we will be able to give you the extra space this time, but please try not to add substantially to the manuscript length in the final revision, if you can help it.

Our final file is showing 32,204 characters, not including spaces (938 abstract; 4,491 introduction; 16,464 results; 9,823 discussion; 488 acknowledgements).

Points 2-8, 10, 11, and 13 have been addressed.

9) eTOC summary: A ~40-50 word summary that describes the context and significance of the findings for a general readership should be included on the title page. We realize that you already provided one in our online system, but we recommend that the statement be written in the present tense and refer to the work in the third person. It should contain "First author name(s) et al..." (or, in your case, "Doan and Monk...") to match our preferred style.

Doan and Monk employ a combination of zebrafish and mouse models in development, adulthood, and injury to demonstrate that the guanine nucleotide exchange factor Dock1 functions with Rac1 in Schwann cells to regulate radial sorting and proper repair.

12) ORCID IDs: ORCID IDs are unique identifiers allowing researchers to create a record of their various scholarly contributions in a single place. Please note that ORCID IDs are now **required** for all authors. At resubmission of your final files, please be sure to provide your ORCID ID and those of all co-authors.

Ryan Doan: 0000-0002-6944-3921

Kelly Monk: 0000-0003-1803-3495